# Galectin-9 binds IgM-BCR to regulate B cell signaling

Anh Cao [1], Nouf Alluqmani[2], Fatima Hifza Mohammed Buhari[3], Laabiah Wasim[1], Logan K. Smith[1], Andrew T. Quaile [4], Michael Shannon[5], Zaki Hakim[2], Hossai Furmli[3], Dylan M. Owen[5], Alexei Savchenko[4,6] & Bebhinn Treanor [1,2,3]

The galectin family of secreted lectins have emerged as important regulators of immune cell function; however, their role in B-cell responses is poorly understood. Here we identify IgM-BCR as a ligand for galectin-9. Furthermore, we show enhanced BCR microcluster formation and signaling in galectin-9-deficient B cells. Notably, treatment with exogenous recombinant galectin-9 nearly completely abolishes BCR signaling. We investigated the molecular mechanism for galectin-9-mediated inhibition of BCR signaling using super-resolution imaging and single-particle tracking. We show that galectin-9 merges pre-existing nanoclusters of IgM-BCR, immobilizes IgM-BCR, and relocalizes IgM-BCR together with the inhibitory molecules CD45 and CD22. In resting naive cells, we use dual-color super-resolution imaging to demonstrate that galectin-9 mediates the close association of IgM and CD22, and propose that the loss of this association provides a mechanism for enhanced activation of galectin-9-deficient B cells.

[1] Department of Immunology, University of Toronto, 1 King's College Circle, Toronto, ON M5S 1A8, Canada. [2] Department of Cell and Systems Biology, University of Toronto, 24 Harbord Street, Toronto, ON M5S 3G5, Canada. [3] Department of Biological Sciences, University of Toronto Scarborough, 1265 Military Trail, Toronto, ON M1C 1A4, Canada. [4] Department of Chemical Engineering and Applied Chemistry, University of Toronto, 200 College Street, Toronto, ON M5S 3E5, Canada. [5] Department of Physics and Randall Division of Cell and Molecular Biophysics, New Hunt's House, King's College London Guy's Campus, London SE1 1UL, UK. [6] Department of Microbiology, Immunology and Infectious Diseases, Cumming School of Medicine, Health Research Innovation Centre 4AA06, University of Calgary, 3330 Hospital Drive NW, Calgary, AB T2N 4N1, Canada. These authors contributed equally: Anh Cao, Nouf Alluqmani, Fatima Hifza Mohammed Buhari, Laabiah Wasim. Correspondence and requests for materials should be addressed to B.T. (eamil: bebhinn.treanor@utoronto.ca)

B cells play a critical role in the immune response and production of protective antibodies. B-cell activation is triggered by binding of antigen to the B-cell receptor (BCR), which initiates a cascade of intracellular signaling through assembly of a multiprotein complex of kinases and adaptors[1]. B-cell activation is accompanied by formation of numerous signaling microclusters[2]. Similar microstructures of antigen receptors have been described in T cells[3] and thus have been proposed to represent the basic unit of lymphocyte signaling[4]. These observations implicate receptor clustering as a mechanism to regulate signaling events, and consequently the cellular outcome of receptor engagement. Indeed, the size and spatial patterning of signaling assemblies significantly contribute to cellular outcomes, with even small variations resulting in altered responses[5–7].

Two key parameters influencing the assembly of signaling clusters and regulation of membrane receptor activation are the constitutive nanoscale clustering of membrane proteins referred to as nanoclusters or protein islands[8–10], and the cell surface mobility of membrane proteins (or nanoclusters of proteins)[7,11,12]. These parameters have important implications for receptor triggering and the assembly of signaling complexes as they influence the interaction between protein partners. Several mechanisms have been identified that impact on the organization and mobility of membrane proteins, including the actin cytoskeleton[11–13], protein–protein interactions[9,14–16], and membrane microdomains defined by lipid composition[8,17].

An often overlooked mechanism controlling membrane protein organization and mobility is the interaction of these cell surface glycoproteins with the family of soluble secreted lectins, known as galectins, which bind and crosslink cell surface proteins, generating glycan-based domains[18]. Indeed, the galectin lattice influences glycoprotein compartmentalization and lateral mobility at the cell surface[19–21]. These proteins have emerged as important regulators of the immune response. For example, T cells from mice deficient in *Mgat5*, which encodes a glycosyltransferase involved in generation of galectin-binding epitopes, have enhanced T-cell receptor (TCR) clustering at the immunological synapse and increased TCR signaling[22]. In comparison to T cells, relatively little is known about glycan–galectin interactions in B cells; however, pre-BCR clustering at the pre-B-cell-stromal cell synapse is dependent on pre-BCR/galectin-1/integrin interactions[23]. Interestingly, a recent study demonstrated decreased antibody titers upon immunization in mice treated with recombinant galectin-9[24]. Galectin-9 belongs to the tandem-repeat subfamily of galectins, which contain two different carbohydrate recognition domains (CRDs) separated by a flexible linker[25]. In T cells, galectin-9 induces cell death of T helper type 1 ($T_H1$) cells through binding to Tim-3[26], as well as suppresses generation of $T_H17$ cells and promotes induction of $T_{regs}$[27]. However, the role of galectin-9 in B cells, and the molecular mechanism for decreased antibody production in galectin-9-treated mice has not been investigated.

Here we identify IgM-BCR and CD45 as ligands for galectin-9. Further, we show enhanced BCR microcluster formation and signaling in galectin-9-deficient B cells. Notably, treatment with exogenous recombinant galectin-9 (rGal9) nearly completely abolishes BCR signaling. We investigated the molecular mechanism for galectin-9-mediated inhibition of BCR signaling. We show using super-resolution imaging and single-particle tracking that galectin-9 brings together pre-formed BCR nanoclusters and reduces BCR mobility at the cell surface, and consequently attenuates BCR microcluster formation. We further interrogated the effect of galectin-9 on the organization of cell surface proteins and demonstrate that galectin-9 increases the molecular density of IgM, CD45, CD22, and CD19 within this galectin lattice. Using dual-color direct stochastic optical reconstruction microscopy (dSTORM), we show that galectin-9 mediates close association of IgM and CD22 in resting naive B cells and thus propose a novel role for galectin-9 in regulation of B-cell activation.

## Results

**Galectin-9 is bound to the surface of primary naive B cells**. To investigate the underlying mechanism for decreased antibody production in galectin-9-treated mice[24], we initially asked if galectin-9 is bound to the surface of B cells. Primary naive B cells were isolated from C57BL/6 (wild type; WT) and $Lgals9^{-/-}$ (Gal9-KO) mice, stained with a fluorescently labeled antibody specific for galectin-9 and examined by flow cytometry and confocal microscopy. We found that galectin-9 is bound to the surface of WT B cells (Fig. 1a), organized in discrete puncta (Fig. 1b). To investigate the in vivo expression of galectin-9, we immunostained inguinal lymph nodes to identify subcapsular sinus macrophages (CD169), B cells (B220), and galectin-9. We found that galectin-9 was readily detectable within the B-cell follicle (Fig. 1c).

**Galectin-9 regulates BCR microclusters and signaling**. To investigate the effect of galectin-9 deficiency on BCR microcluster formation we employed artificial planar lipid bilayers in which fluorescently labeled anti-BCR antibodies as surrogate antigen are tethered to mimic the in vivo environment of antigen encounter[28]. Naive B cells from WT and Gal9-KO mice were settled on lipid bilayers containing anti-kappa, fixed after 90 s, and visualized by confocal microscopy. WT B cells spread and form multiple BCR-antigen microclusters[2,29] (Fig. 2a). Gal9-KO B cells also spread and form microclusters; however, the area of cell-bilayer contact, and total intensity of antigen is increased (Fig. 2a–c). Moreover, the mean antigen intensity is also increased in Gal9-KO B cells, indicating that the amount of BCR-antigen within individual clusters is increased (Fig. 2d). Given this enhanced microcluster formation, we asked if BCR signaling was altered in Gal9-deficient B cells. Consistent with microcluster data, total tyrosine phosphorylation was increased in Gal9-KO B cells (Fig. 2e). To further investigate this enhanced signaling, we examined phosphorylation of downstream signaling molecules, including CD19, Akt, and extracellular regulated kinase (ERK1/2). Although we did not detect any increase in phospho-CD19 or phospho-Akt, we found that phosphorylation of ERK1/2 was increased in Gal9-KO B cells (Fig. 2f–h). We verified that these differences were not due to altered expression of BCR or CD19 (Supplementary Fig. 1). These data demonstrate that galectin-9 regulates BCR microcluster formation and signaling.

To determine if enhanced microcluster formation and signaling is specifically due to loss of galectin-9 at the cell surface, we titrated the amount of recombinant galectin-9 (rGal9) added to Gal9-KO B cells from 0.1 to 1 μM and then stained cells with anti-galectin-9 antibody and compared this to untreated WT cells. We found that 0.1 μM rGal9 added to Gal9-KO cells was roughly equivalent to the level of endogenous galectin-9 on WT cells (Fig. 3a). We treated Gal9-KO B cells with 0.1 μM rGal9 and then stimulated with anti-IgM and examined ERK phosphorylation as we found this pathway was enhanced in galectin-9-deficient cells. We found that this treatment was sufficient to restore BCR signaling to that observed in WT B cells (Fig. 3b), demonstrating that the effect of deficiency of galectin-9 is indeed due to lack of galectin-9 at the cell surface. Given that galectin family members share the basic glycan ligand, *N*-acetyllactosamine, we examined an alternative galectin-deficient mouse model. In contrast to Gal9-KO B cells, we found no difference in BCR

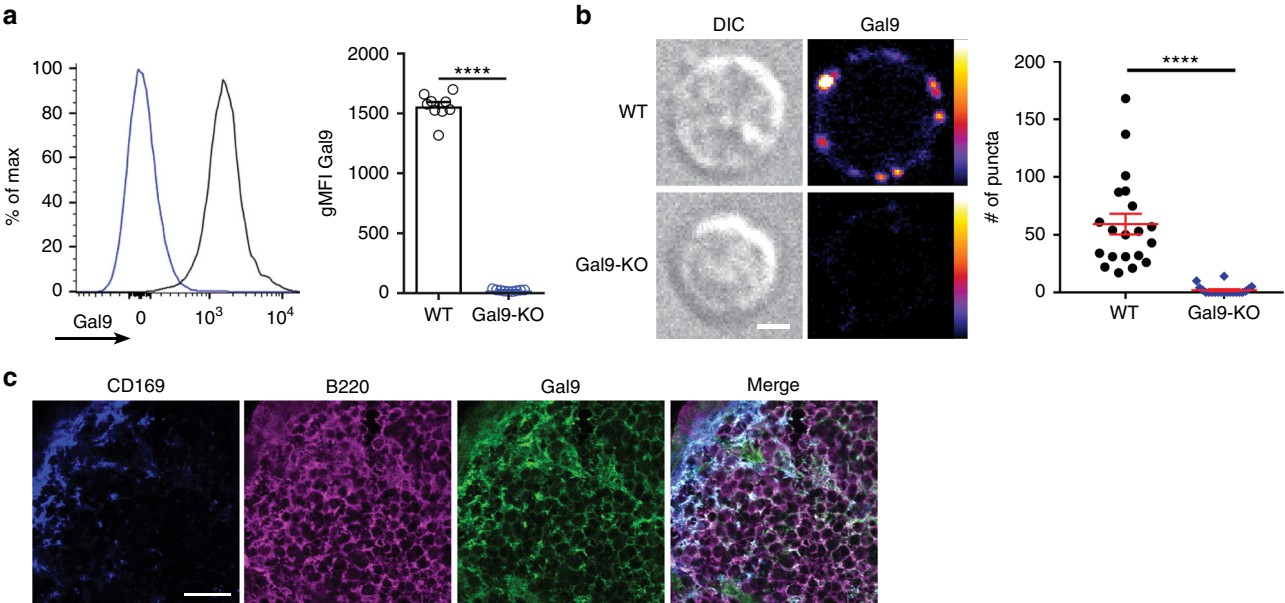

**Fig. 1** Galectin-9 is bound to the surface of primary naive B cells. **a** Representative flow cytometry plot (left) and quantification (right) of geometric mean ± SEM of surface staining for galectin-9 in WT (black) and Gal9-KO (blue) B cells from nine independent experiments. **b** Representative DIC (left) and confocal microscopy images (right) mapped to an 8-bit fire color scale (ImageJ) of primary WT (top) and Gal9-KO B cells (bottom) stained for surface galectin-9. Quantification of number of galectin-9 puncta is shown on the right (each dot represents 1 cell, 20 cells measured per condition) with the mean ± SEM indicated by the red bar. Scale bar 2 μm. Data representative of three independent experiments. **c** Representative confocal microscopy images of cryosections of the inguinal lymph node of WT B cells stained for subcapsular sinus macrophages (CD169; blue), B cells (B220; magenta), and Gal9 (green). Scale bar 20 μm. Data representative of three independent experiments. Statistical significance was assessed by Mann-Whitney, ****$p < 0.0001$

microcluster formation or downstream signaling in galectin-1-deficient B cells (Supplementary Fig. 2). These data, while not exhaustive of the galectin family, provide strong evidence for specificity of galectin-9 in regulation of these early events of B-cell activation, consistent with studies demonstrating that structural differences in the carbohydrate recognition domain (CRD) of galectin-1 and galectin-9 define specific interactions with cell surface glycoproteins[30].

We noted that ligands for galectin-9 at the cell surface were not saturated in primary naive B cells and we could add 10-fold more rGal9 to both WT and Gal9-KO B cells (Fig. 3a). Thus, we asked what happens to BCR signaling if we increase the amount of galectin-9 bound to the surface. We observed a substantial reduction in total tyrosine phosphorylation, as well as phosphorylation of CD19, Akt, and ERK upon pre-treatment of WT cells with rGal9 (Fig. 3c–f). These differences were not due to altered expression of BCR, CD19, or CD45 by rGal9 (Supplementary Fig. 3), nor galectin-9-induced cell death (Supplementary Fig. 4), as has been reported for thymocytes and T cells[26,31,32].

**Galectin-9 binds IgM-BCR and CD45.** Our data demonstrate an important role for galectin-9 in regulating BCR signaling; however, the cell surface glycoproteins that bind galectin-9 in B cells have not been identified. To identify potential ligands we performed a pull-down assay using rGal9 followed by mass spectrometry[33]. To not compete with endogenous galectin-9, Gal9-KO primary B cells were lysed, incubated with FLAG-tagged rGal9 followed by magnetic separation using beads conjugated with anti-FLAG, and the ligands were eluted using lactose (Fig. 4a). Isolated proteins were subjected to in-solution tryptic digestion and liquid chromatography tandem mass spectrometry. We identified several potential ligands of galectin-9 in B cells, including CD45 (*Ptprc*), IgM-BCR (*Ighm*), and the BCR-associated signaling chain CD79b (Igβ) (Table 1). To confirm the identity of these ligands, we performed a pull-down assay

using rGal9 under stringent cell lysis conditions to disrupt protein–protein interactions followed by western blotting (Fig. 4b). Consistent with our mass spectrometry results, we identified CD45 and IgM-BCR as potential ligands of galectin-9. To verify that these were indeed direct ligands of galectin-9, we performed far-western analysis[34] using FLAG-tagged rGal9 as bait protein. We detected galectin-9-binding proteins corresponding to the size of CD45 and IgM in reference blots (Fig. 4c) in primary murine B cells. Importantly, a band corresponding to the size of IgM was not detected in far-western using cell lysates from the A20 B-cell line that does not express endogenous IgM. Conversely, a band corresponding to the size of IgM was detected in lysates of A20 B cells expressing hen egg lysozyme (Hel)-specific IgM (D1.3). Neither CD45 (B220) nor IgM bands were detected when Jurkat T cells were used. These data provide strong evidence that galectin-9 binds IgM-BCR and CD45.

**Galectin-9 alters IgM-BCR nanoclusters.** Identification of IgM-BCR as a ligand of galectin-9, and the potential of galectins to act as cell surface scaffolding proteins[19] led us to ask whether galectin-9 has a role in organizing IgM-BCR nanoclusters. To investigate this, we employed dSTORM[35], which permits a lateral resolution of approximately 20 nm, and has been used to study the organization of IgM, IgD, and IgG BCRs[9,10,36]. Primary WT and Gal9-KO B cells were labeled with Alexa Fluor 647-conjugated anti-IgM Fab fragment, settled on non-BCR-stimulating anti-major histocompatibility complex (MHC) II-coated coverslips to adhere cells, then fixed and imaged by total internal reflection fluorescence microscopy (TIRFM). IgM-BCR are organized into nanoscale clusters in WT B cells[9] (Fig. 5a). Nanoclusters of IgM-BCR were also clearly visible in dSTORM images in Gal9-KO B cells, with no obvious difference compared to WT B cells (Fig. 5a). To quantify clustering tendency of IgM-BCR, we used two methods commonly used to assess distribution of proteins from super-resolution images[37]. The

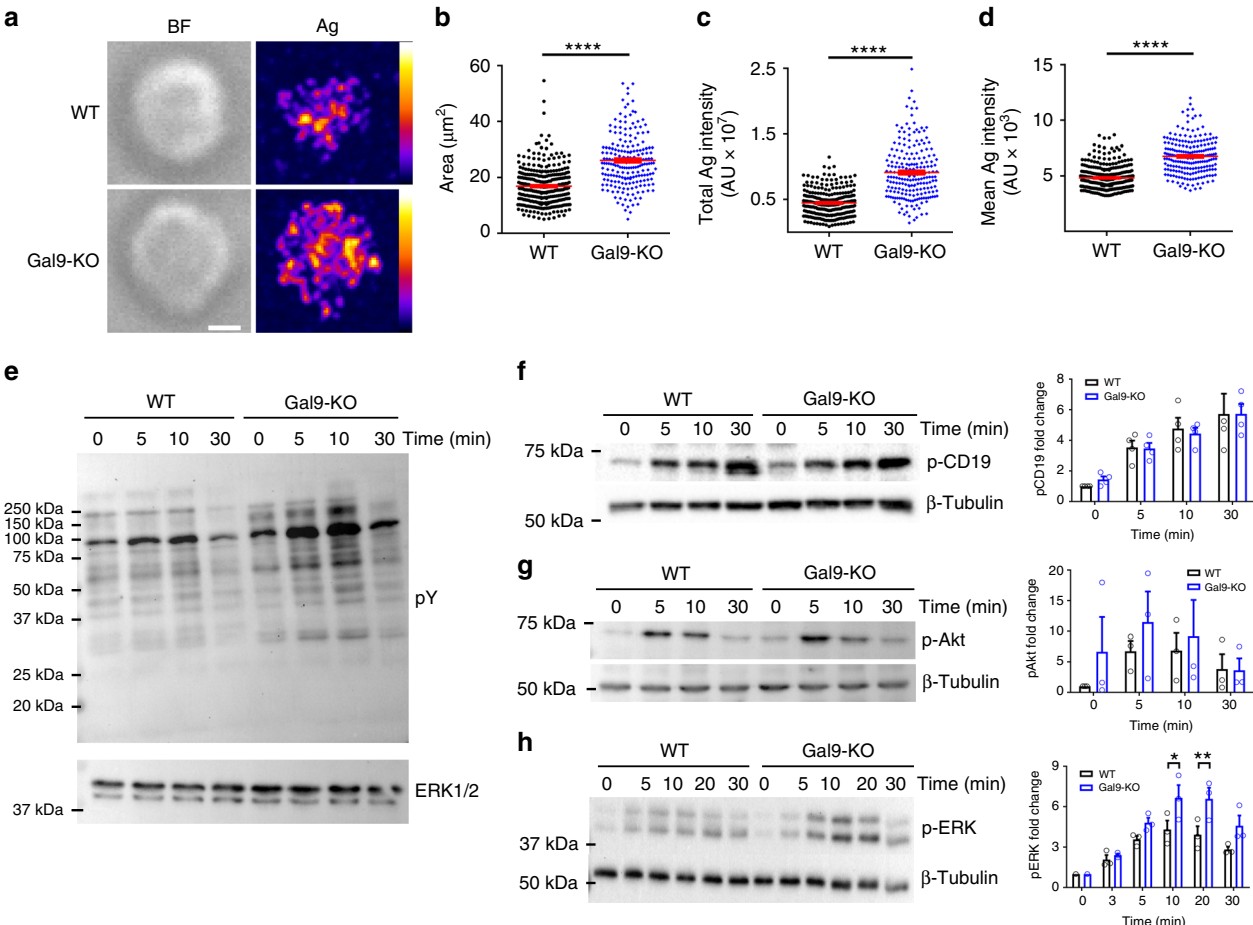

**Fig. 2** Galectin-9 regulates BCR microcluster formation and signaling. **a** Representative images of primary naive WT (top) and Gal9-KO B cells (bottom) fixed on bilayers containing anti-kappa as surrogate antigen (Ag) after 90 s of spreading and imaged by confocal microscopy. Brightfield (left) and confocal (right) visualizing antigen mapped to an 8-bit fire color scale (ImageJ). Scale bar 2 μm. Quantification of **b** area of spreading, **c** total antigen fluorescence intensity at the cell-bilayer contact, and **d** mean intensity of antigen for WT (black circles) and Gal9-KO (blue diamonds) cells (each dot represents 1 cell, 200 cells measured per condition), with the mean ± SEM indicated by the red bar, ****$p < 0.0001$, Mann-Whitney test. Data representative of at least three independent experiments. **e–h** Primary naive B cells from WT and Gal9-KO mice were settled onto anti-IgM-coated plates for the indicated time. Cells were lysed and subjected to SDS-PAGE followed by immunoblotting with **e** anti-phosphotyrosine and anti-ERK1/2, **f** anti-phospho-CD19, **g** anti-phospho-Akt, and **h** anti-phospho ERK1/2 (pERK) and anti-β tubulin. Data representative of at least three independent experiments. **f** Quantification of the fold increase in pCD19, pAkt, and pERK, with the mean ± SEM indicated by bar. Data were analyzed by two-way ANOVA, followed by Sidak's multiple comparisons test; *$p < 0.05$, **$p < 0.01$

Hopkins index, which evaluates clustering tendency compared to a random distribution (value of 0.5), was 0.75 for IgM-BCR on both WT and Gal9-KO B cells (Fig. 5b). The $H$ function derived from Ripley's $K$ function evaluates the extent of clustering; distance of the $H$ function peak is related to cluster radius and peak height depends on density of molecules in clusters. We found no difference in the $H$ function curve in Gal9-KO B cells compared to WT B cells. These findings suggest that galectin-9 does not mediate formation of IgM-BCR nanoclusters; however, galectin-9 is sparsely distributed at the cell surface of naive B cells (Fig. 1b), and therefore dSTORM analysis based on randomly selected regions may underestimate an effect specifically within the galectin-9 lattice. To focus our dSTORM analysis on the galectin-9 lattice, we treated Gal9-KO B cells with fluorescently labeled rGal9. We found that this treatment altered the organization of IgM-BCR, which appeared more clustered compared to WT and Gal9-KO cells (Fig. 5a). Both the Hopkin's index and the $H$ function of Ripley's $K$ derived from regions where galectin-9 localized indicated that IgM-BCR was more highly clustered (Fig. 5b, c). The radius of clusters of IgM-BCR inside the galectin-9 lattice is approximately 150–250 nm, compared to 50–100 nm

in WT and Gal9-KO B cells (Fig. 5c). To further quantify the effect of galectin-9 on organization of IgM, we used Bayesian cluster analysis[38]. This method is advantageous because: (1) it accounts for non-negligible unclustered background, which can interfere with Ripley's and Hopkins analysis; (2) it takes account of the uncertainty associated with each localization assigned by the Gaussian function; and (3) it reduces the bias arising from subjective user-defined analysis parameters, such as threshold and cluster radius. Using this method, we identified the number of clusters per region, cluster radii, number of molecules per cluster, and percentage of localizations in clusters from $3 \times 3$ μm regions of interest from reconstructed images. Treatment with rGal9 decreased the number of clusters (Fig. 5d), whereas radius of clusters (Fig. 5e) and number of molecules per cluster (Fig. 5f) increased. There was no difference in percentage of localizations within clusters between WT, Gal9-KO, and Gal9-KO treated with rGal9 (Fig. 5g). These findings suggest that galectin-9 brings pre-existing IgM-BCR nanoclusters together, increasing size of clusters and number of molecules per cluster, but without affecting percentage of molecules residing in clusters.

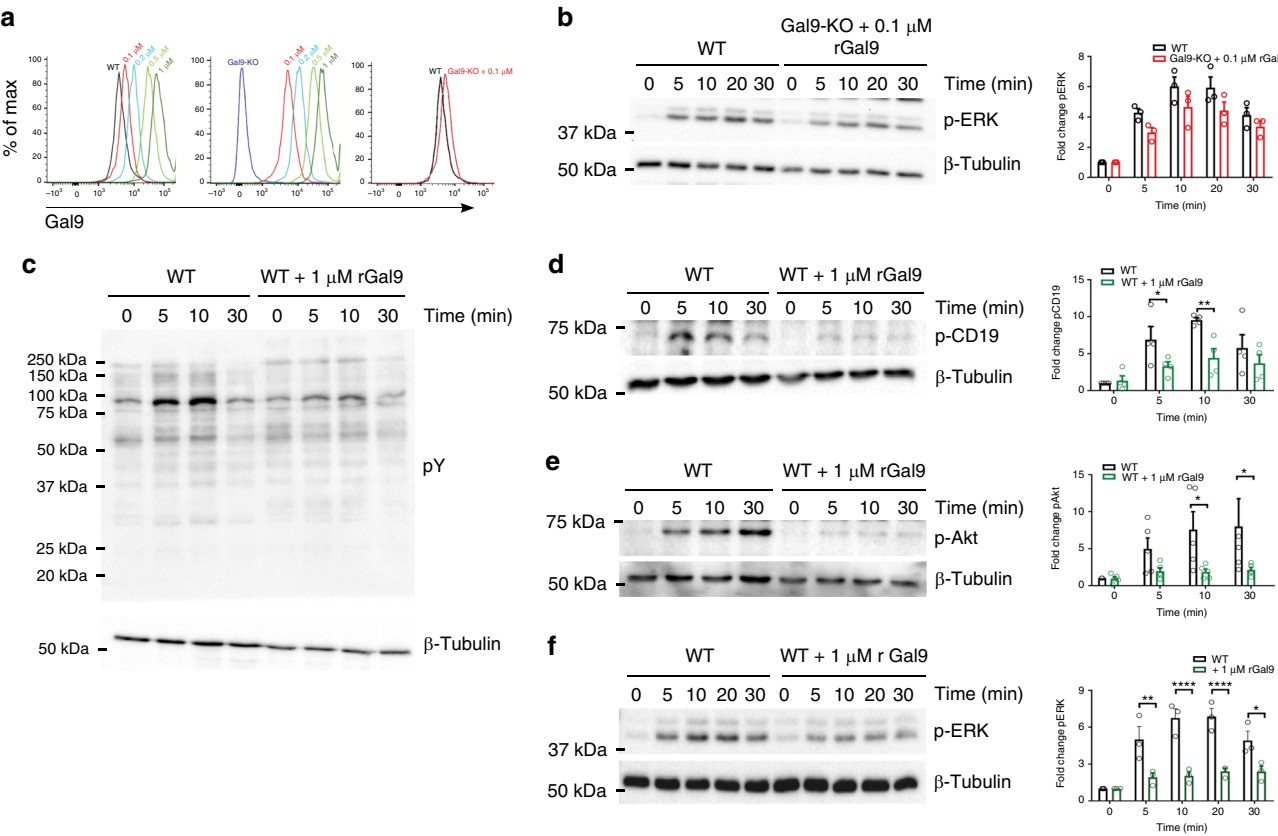

**Fig. 3** Treatment with exogenous galectin-9 suppresses BCR signaling. **a** Representative flow cytometric histograms of WT (left) and Gal9-KO (middle) B cells either untreated or treated with various concentrations of recombinant galectin-9 (rGal9; 0.1, 0.2, 0.5, and 1 μM) followed by surface staining for galectin-9 and analyzed using flow cytometry. Overlay of endogenous galectin-9 surface expression in WT cells, and Gal9-KO cells treated with 0.1 μM rGal9 (right). **b** Naive B cells from WT and Gal9-KO mice treated with 0.1 μM rGal9 were settled onto anti-IgM-coated plates for the indicated time. Cells were lysed and subjected to SDS-PAGE followed by immunoblotting with anti-phospho ERK1/2 and anti-β tubulin (left panel). Quantification of the fold change in pERK over time, averaged over two independent experiments with the mean ± SEM indicated by the bar (right panel). **c–f** Naive B cells from WT mice were treated with 1 μM rGal9 and settled onto anti-IgM-coated plates for the indicated time. Cells were lysed and subjected to SDS-PAGE followed by immunoblotting with **c** anti-phosphotyrosine and ERK1/2, **d** anti-phospho-CD19, **e** anti-phospho-Akt, and **f** anti-phospho ERK1/2 and anti-β tubulin. Quantification of the fold change in pCD19, pAkt, and pERK over time, averaged over three independent experiments, with the mean ± SEM indicated by the bar is shown in the right panel. Statistical significance measure by two-way ANOVA followed by Sidak's multiple comparisons test; ****$p < 0.0001$, **$p < 0.01$, *$p < 0.05$

**Galectin-9 immobilizes IgM-BCR**. The mobility of IgM-BCR is a critical factor regulating BCR signaling[9,12]. Given dSTORM analysis showed galectin-9 increased cluster size, we wondered whether altered BCR mobility may account for suppressed BCR signaling upon treatment with rGal9. To investigate the effect of galectin-9 on IgM mobility in the steady state, we labeled single particles of IgM on WT and Gal9-KO B cells with Atto-633-conjugated anti-IgM Fab fragments. Cells were settled on anti-MHC II-coated coverslips and single molecules of IgM were visualized using TIRFM. The median diffusion coefficient of IgM-BCR on Gal9-KO B cells was approximately 30% higher than that on WT B cells (0.037 μm²/s compared to 0.027 μm²/s; Fig. 6a), consistent with a decrease in the relative frequency of slow-moving IgM-BCR in Gal9-KO B cells (Fig. 6b). To examine the mobility of IgM-BCR inside the galectin-9 lattice, we treated Gal9-KO B cells with fluorescently labeled rGal9 and visualized and tracked IgM-BCR. We defined regions of high and low galectin-9 using the fluorescence intensity of galectin-9 and then calculated diffusion coefficient in these regions (Fig. 6c). The median diffusion coefficient of IgM in galectin-9-high regions was reduced twofold compared to galectin-9-low regions (0.012 μm²/s compared to 0.028 μm²/s) (Fig. 6d). Approximately 65% of tracked IgM molecules are largely immobilized inside the mask

compared to 45% of tracked IgM molecules outside the mask (Fig. 6e). This finding provides direct evidence for galectin-9 to restrict the mobility of IgM molecules on the B-cell membrane.

We hypothesized that galectin-9-mediated immobilization of IgM-BCR would reduce BCR localization to the contact with an antigen-containing membrane. To test this, we settled WT B cells or cells pre-treated with rGal9 on planar lipid bilayers containing anti-kappa as surrogate antigen for 90 s and then fixed and visualized by confocal microscopy. We found that BCR-antigen aggregation is reduced in cells treated with rGal9 (Fig. 6f, g). These findings suggest that galectin-9 binds IgM-BCR inducing aggregation and immobilization of IgM nanoclusters, which consequently affects the formation of signaling microclusters upon BCR stimulation.

**Galectin-9 increases the density of IgM-BCR and co-receptors**. Our finding that rGal9 induces an increase in the size of IgM-BCR clusters and immobilization of IgM are also features consistent with the activation of B cells[9,10,15,39]. To examine if galectin-9 mediated clustering of IgM was sufficient to induce signaling, we examined calcium flux upon treatment of cells with rGal9 by flow cytometry. In contrast to stimulation with anti-BCR antibodies, we did not detect any increase in intracellular

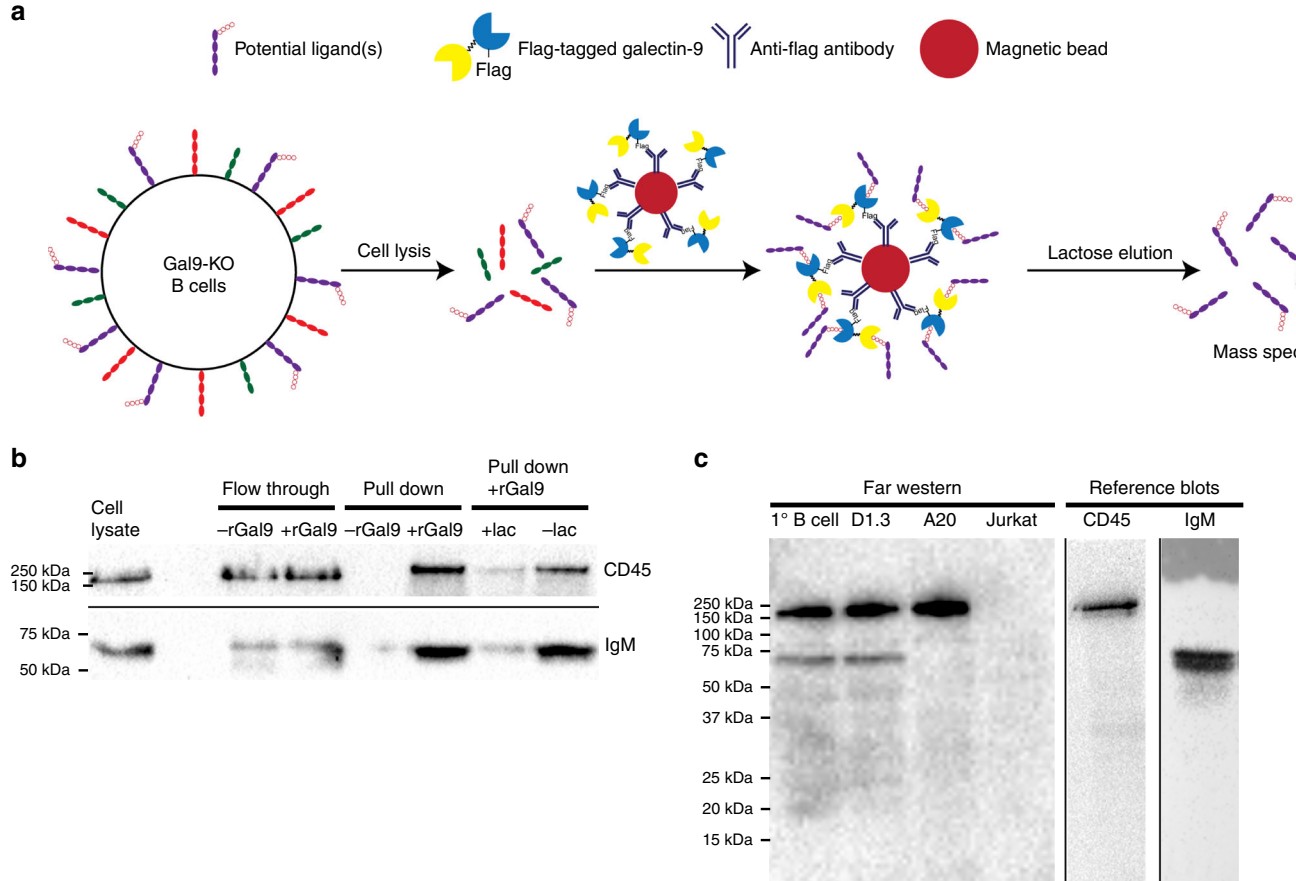

**Fig. 4** Galectin-9 binds IgM-BCR and CD45. **a** Schematic diagram of pull-down assay using FLAG-tagged rGal9 coupled to magnetic beads. Galectin-9-interacting ligands were eluted by treatment with lactose and then identified by mass spectrometry. **b** Lactose eluted proteins from beads coated (+rGal9) or not (−rGal9) with rGal9, or in the presence of lactose (+lac) or not (−lac) were subjected to SDS-PAGE following by immunoblotting using antibodies specific to CD45 and IgM. **c** Cell lysates from primary murine B cells (lane 1), A20 B cells expressing Hel-specific IgM (D1.3; lane 2), A20 B cells (lane 3), and Jurkat T cells (lane 4) were lysed and subjected to far-western analysis using FLAG-tagged rGal9. Reference blots probed for CD45 and IgM are shown on the right. Data representative of three independent experiments

calcium upon addition of rGal9 (Supplementary Fig. 5). This finding raises the question of why treatment with rGal9, which clusters BCR, does not itself induce B-cell activation. Our data also identified CD45, the most abundant phosphatase on the B-cell surface, as a ligand of galectin-9 (Fig. 4). CD45 is a member of the protein tyrosine phosphatase family and plays an important role in both positive and negative regulation of B-cell activation through dephosphorylation of the activating and inhibitory sites of the Src family kinase Lyn[40]. This dual regulation of BCR signaling by CD45 is dependent on the localization of CD45, as CD45 constitutively dephosphorylates and inactivates Lyn within glycosphingolipid-enriched domains (GEMs), and BCR ligation induces sequestration of CD45 from GEMs[41]. We predicted that galectin-9 modifies the spatial organization of IgM and CD45 to attenuate signaling. To visualize the effect of galectin-9 on organization of CD45 relative to IgM-BCR, we treated primary WT B cells with 1 μM rGal9 and then fixed and immunostained using anti-CD45, anti-IgM, and anti-galectin-9 antibodies, and imaged by confocal microscopy. Interestingly, rGal9 often forms a cap at one side of the cell, consistent with the ability of galectins to form a lattice-like structure through galectin–glycoprotein interactions (Fig. 7a). We also observed that within the galectin-9 cap, the fluorescence intensity of CD45 and IgM is increased (Fig. 7a, b). To quantify this observation, we developed a protocol to define galectin-9-high regions (Gal9high) and galectin-9-low regions (Gal9low) (Fig. 7c) and found that the mean fluorescence intensity

of both CD45 and IgM is significantly higher in Gal9high regions compared to Gal9low regions (Fig. 7d).

Reorganization of IgM and CD45 is not due to a global change in membrane structure, as the intensity of IgD, an isotype of BCR that is expressed 10-fold higher than IgM[9], was not increased within the galectin-9 lattice (Supplementary Fig. 6a–c).

One of the best characterized inhibitory co-receptors expressed on B cells is CD22[42]. The extracellular domain of CD22 is also glycosylated and contains a sialic acid-binding domain, which binds α2,6-linked sialic acids and mediates homotypic interactions between CD22 molecules and heterotypic interactions with other glycosylated proteins, including CD45 and IgM[43,44]. Thus, we investigated if rGal9 also alters organization of CD22. We observed clusters of CD22 with higher fluorescence intensity inside the galectin-9 cap, which colocalized with IgM clusters (Fig. 7e, f). Consistent with this, the mean fluorescence intensity of CD22 is increased inside Gal9high regions (Fig. 7g). These findings suggest that rGal9 increases the density of CD22, coincident with the enrichment of IgM, within the galectin-9 lattice. Again, to confirm the specificity of CD22 enrichment within the galectin-9 lattice, we examined localization of inhibitory receptor FcγRIIb, upon treatment with rGal9, and found no enrichment within Gal9high regions (Supplementary Fig. 6d–f).

Our findings suggest that galectin-9 enhances the interaction between IgM and CD22 and may provide a mechanistic basis for

**Table 1 Summary table of top hits of galectin-9-binding proteins identified by MS**

| UniprotKB ID | Gene | Log(e) | Log(l) | % Measured | % Corrected | Unique peptides | Total peptides | Mr | Total peptides/ molecular weight |
|---|---|---|---|---|---|---|---|---|---|
| Q8VDN2 | Atp1a1 | −165.4 | 5.7 | 17 | 31 | 12 | 29 | 112.9 | 0.256864482 |
| Q8C129 | Lnpep | −119.5 | 5.58 | 11 | 23 | 10 | 21 | 117.2 | 0.179180887 |
| P06800 | Ptprc | −114.7 | 5.72 | 9.2 | 17 | 11 | 25 | 144.5 | 0.173010381 |
| P24668 | M6pr | −75.3 | 5.8 | 21 | 36 | 5 | 16 | 31.2 | 0.512820513 |
| Q03265 | Atp5a1 | −70.2 | 5.33 | 18 | 24 | 7 | 12 | 59.7 | 0.201005025 |
| P11835 | Itgb2 | −63.2 | 4.98 | 6.8 | 9 | 4 | 7 | 84.8 | 0.08254717 |
| P19437 | Ms4a1 | −60.9 | 5.71 | 22 | 60 | 5 | 25 | 31.9 | 0.78369906 |
| P10852 | Slc3a2 | −59.2 | 5.34 | 13 | 21 | 5 | 12 | 62.2 | 0.192926045 |
| Q8BG07 | Pld4 | −52.3 | 5.3 | 16 | 32 | 5 | 11 | 56.1 | 0.196078431 |
| P01872 | Ighm | −39.3 | 5.3 | 8.6 | 16 | 3 | 9 | 50 | 0.18 |
| Q62192 | Cd180 | −36.1 | 5.19 | 6.7 | 14 | 4 | 9 | 74.3 | 0.121130552 |
| P51881 | Slc25a5 | −32.7 | 5.77 | 7 | 10 | 3 | 8 | 32.9 | 0.243161094 |
| Q61735 | Cd47 | −25.6 | 5.39 | 7.7 | 26 | 2 | 11 | 35.3 | 0.311614731 |
| P15530 | Cd79b | −18.8 | 5.11 | 7.6 | 22 | 2 | 7 | 32.1 | 0.218068536 |
| O35424 | H2-Ob | −15.4 | 5.18 | 6.6 | 10 | 2 | 6 | 30.4 | 0.197368421 |

Relevant statistics from one replicate experiment. Top hits here identified using the following criteria: log(e) score ≤ −10; total peptides ≥ 5; unique peptides ≥ 2 in at least two independent experiments. Hits observed in "no bait" controls were removed. Data acquired with input of 2e7 cells and 200 mM lactose elution

the inhibitory effect of galectin-9 on B-cell activation. The intracellular domain of CD22 contains three immunoreceptor tyrosine-based inhibitory motifs that are phosphorylated upon BCR ligation[43] and recruit other phosphatases such as Src homology region 2 domain-containing phosphatase-1 (SHP-1), which dephosphorylates Syk, PLCγ2, and CD19 to dampen BCR signaling[45]. Given our finding of reduced phosphorylation of CD19 in BCR stimulated cells pre-treated with rGal9 (Fig. 3d), we examined if CD19 is also localized within the galectin-9 lattice, and found that CD19 is indeed enriched within Gal9[high] regions (Fig. 7h–j). These findings suggest that galectin-9 reorganizes IgM-BCR and its co-receptor CD19 together with inhibitory co-receptors CD45 and CD22 to suppress BCR signaling.

GEMs (or lipid rafts) play an important role in initiation of BCR signaling, and the localization of proteins within these domains have important consequences for activation of signaling[46]. We found that when B cells are treated with rGal9, galectin-9 often forms a cap on one side of the cell, reminiscent of the polarization of lipid rafts upon lymphocyte activation[47,48]. Thus, we asked whether galectin-9 alters the distribution of lipid rafts and association of CD45 and CD22 with these domains. We treated primary B cells with rGal9 and then labeled lipid rafts with fluorescently conjugated cholera toxin β-subunit (CT-B) and immunostained for galectin-9, CD45, and CD22. Treatment with rGal9 increased the intensity of CT-B within galectin-9 regions and this region was enriched for CD45 and CD22 (Fig. 8a, b, e, f). We generated masks defining CT-B-high (lipid raft high, LR[high]) and CT-B-low (lipid raft-low regions, LR[low]) regions and then examined the intensity of CD45 and CD22 within these regions (Fig. 8c). The mean intensity of both CD45 and CD22 is increased within lipid raft-rich regions upon treatment with rGal9 (Fig. 8d, g). Together with our previous findings, these data suggest that treatment with rGal9 induces coalescence of lipid raft domains containing CD45 and CD22 to suppress CD19 and BCR activation.

**Galectin-9 mediates association of IgM and CD22.** We used rGal9 as a tool to understand the molecular mechanism for enhanced B-cell activation in galectin-9-deficient B cells. These studies revealed that rGal9 alters the organization of CD45 and CD22 with respect to IgM-BCR (Fig. 7) and this correlates with suppressed BCR signaling (Fig. 3). To investigate if lack of galectin-9 altered association of IgM-BCR with these proteins in

the steady state, we employed dual-color dSTORM to examine colocalization of CD45 and CD22 with IgM-BCR in WT and Gal9-KO B cells. We validated our dual-color dSTORM imaging and analysis parameters by labeling IgM with both anti-IgM coupled to Alexa 488 and Alexa 647 (Supplementary Fig. 7). We labeled primary naive B cells from WT and Gal9-KO mice with Alexa 488-labeled IgM-specific Fab fragment and Alexa 647-labeled anti-CD45 (B220) or anti-CD22 and settled them on anti-MHC II-coated coverslips. Dual-color dSTORM revealed that CD45 is localized in nanoscale clusters, which minimally overlap with IgM clusters in WT B cells, with no obvious difference compared to Gal9-KO cells (Fig. 9a). We also found no difference in the Hopkins index, H function curve, mean cluster diameter, and mean cluster area of CD45 in WT and Gal9-KO B cells (Fig. 9b–e). To examine colocalization of IgM and CD45 we performed coordinate-based colocalization (CBC) analysis[49], which ranges from −1 (perfectly segregated) to +1 (perfectly colocalized). CBC analysis of WT cells revealed a distribution histogram with two peaks, one approaching −1 and one approaching +1, indicating some degree of colocalization between IgM and CD45 in the resting state; however, the median CBC value for IgM and CD45 was 0.018 indicating that across the population of molecules the distribution is largely uncorrelated (Fig. 9f). We observed a similar distribution of correlation coefficients and nearest-neighbor distance in Gal9-KO B cells (Fig. 9f, g), suggesting that galectin-9 does not influence the association of IgM and CD45 to a significant degree in the steady state.

In contrast to CD45, dual-color dSTORM of IgM and CD22 showed a larger degree of colocalization based on visual inspection of dSTORM images (Fig. 9h). Consistent with previous reports[50] CD22 is highly clustered on the surface of resting primary naive B cells. We noted a small but statistically insignificant difference in both the mean Hopkin's index and the H function curve, with CD22 appearing slightly less clustered in Gal9-KO B cells compared to WT cells (Fig. 9i, j). However, we noted no difference in mean cluster diameter or cluster area of CD22 between WT and Gal9-KO B cells (Fig. 9k, l). CBC analysis of colocalization of IgM and CD22 revealed that the frequency distribution was right shifted in WT B cells, indicating that CD22 is more colocalized with IgM than CD45 in the steady state (Fig. 9m). Indeed, the median CBC value in WT cells was 0.22. In contrast, the median CBC value in Gal9-KO B cells was 0.12, and the nearest-neighbor analysis revealed that the mean distance between IgM and CD22 molecules was increased (Fig. 9m, n).

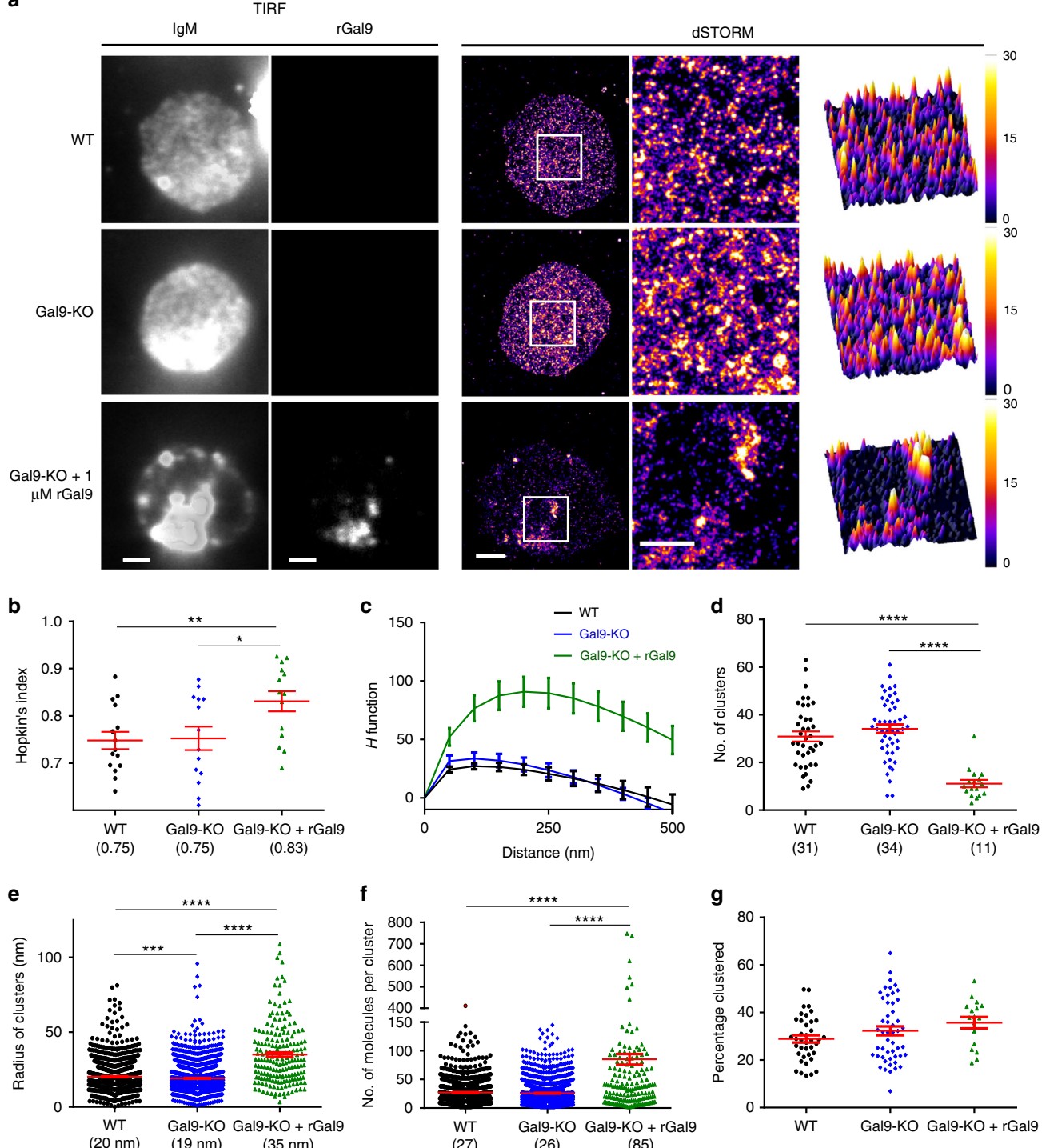

**Fig. 5** Galectin-9 alters IgM-BCR nanoclusters. **a** TIRFM image of surface IgM and fluorescently labeled rGal9 before bleaching for image acquisition (two left panels respectively). dSTORM images reconstructed from single-molecule localization processed by Thunderstorm software mapped to a fire color scale as indicated; the magnified region (3 × 3 μm) from ROI (white box) is shown as 2D image (middle) and 3D surface plot (right) in the order of WT (top), Gal9-KO (middle), and Gal9-KO + 1 μM rGal9 (bottom). Scale bar represents 2 μm. **b** Quantification of the distribution of IgM by *H* function and **c** Hopkins index of localizations inside ROIs. **d**–**g** Reconstructed images were analyzed by a model-based Bayesian approach to identify nanoclusters and their physical properties. **d** Number of clusters (one point per ROI). **e** Cluster radii (one point per cluster). **f** Number of molecules (one point per cluster). **g** Percentage of localization in clusters (one point per ROI). Each category contains at least 15 ROIs from three independent experiments (at least four cells per experiment). Statistical analysis was performed using Kruskal-Wallis test with Dunn's multiple comparison test (**d**, **e**, **f**) and one-way ANOVA with Tukey's multiple comparison test (**b**, **g**). Red bars indicate mean ± SEM; *$p < 0.05$, **$p < 0.01$, ***$p < 0.001$, ****$p < 0.0001$

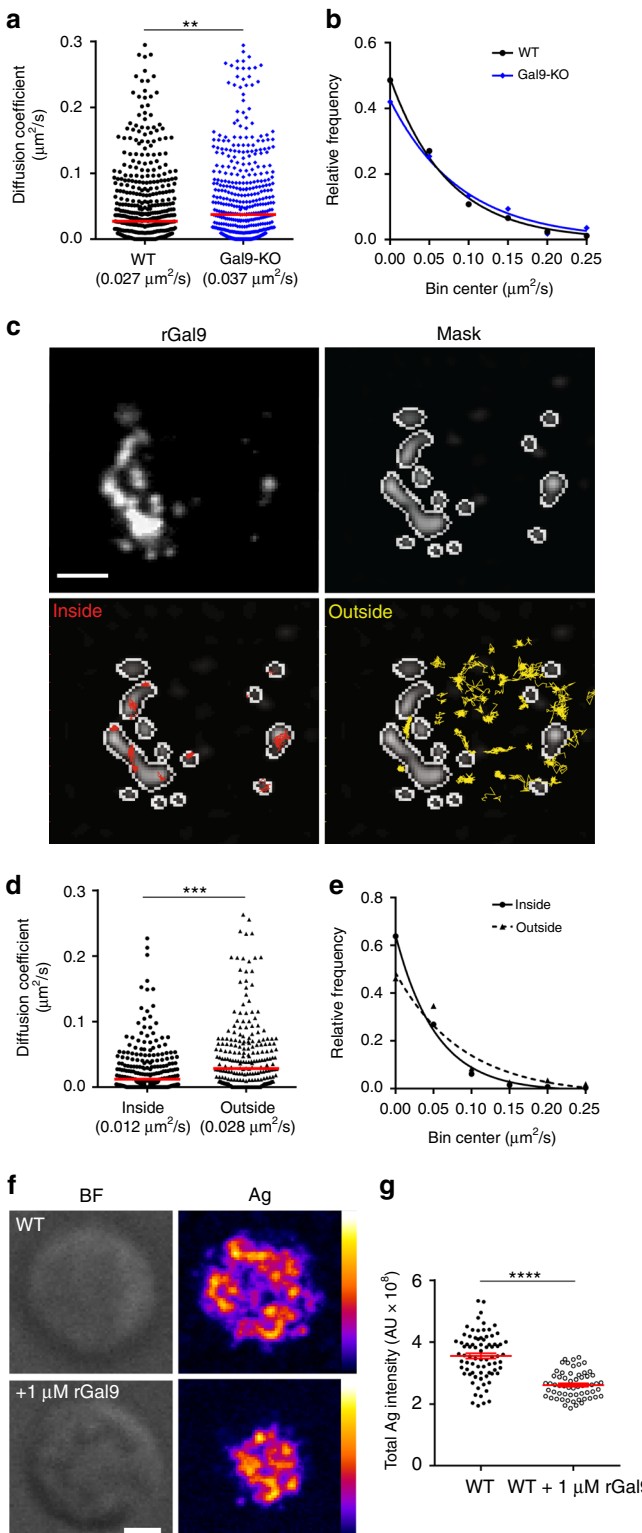

**Fig. 6** Galectin-9 immobilizes IgM-BCR and attenuates BCR microclustering. **a** Diffusion coefficients and **b** frequency distribution histogram of single-particle tracking of IgM in WT (black circle) or Gal9-KO (blue diamond) primary B cells with the median indicated in red. Five hundred representative diffusion coefficients from a total of at least 1500 tracks from three independent experiments. **c** Representative TIRF image of fluorescently labeled rGal9 on primary B cell (left) and mask (right) created to differentiate tracks inside Gal9 regions (lower left, red lines) and tracks outside Gal9 regions (lower right, yellow lines). **d** Diffusion coefficients and **e** frequency distribution inside Gal9 regions (black circle) and outside Gal9 regions (black triangle) with the median indicated in red. In all, 250 representative diffusion coefficients from a total of at least 900 tracks from three independent experiments. **f** Representative TIRF microscopy images of WT cell (top) and WT cells treated with 1 μM rGal9 (bottom) on artificial planar lipid bilayers containing anti-kappa mapped to an 8-bit fire color scale (ImageJ). **g** Quantification of the total antigen intensity at the cell-bilayer interface in WT (black circles) and WT treated with rGal9 (open circles), with the mean ± SEM indicated by the red bar. Scale bar represents 2 μm. Statistical significance assessed by Mann-Whitney; $****p < 0.0001$, $***p < 0.001$, $**p < 0.01$

we show that the ligands of galectin-9 at the cell surface are not saturated in freshly isolated naive B cells and treatment with rGal9 nearly abolishes BCR signaling. Using a pull-down assay and mass spectrometry we identified IgM and CD45 as ligands of galectin-9 in primary B cells. We further show that galectin-9 plays a role in regulating the spatial organization and dynamics of IgM-BCR and its localization with regulatory surface proteins. We propose that galectin-9 organizes IgM-BCR into larger clusters that restrict the mobility of IgM and relocalizes inhibitory molecules, including CD45 and CD22 to directly inhibit BCR signaling (Fig. 10).

The spatial organization of cell surface proteins in constitutive nanoscale clusters has emerged as a general principle of cell surface proteins[51]. However, the molecular mechanism regulating the size, composition, and stability of these assemblies is an open question. The formation and stability of IgM-BCR nanoclusters is not dependent on the actin cortex, as depolymerization of actin does not alter the size or density of IgM-BCR nanoclusters[9]. Thus, we hypothesized that the galectin-9 lattice may play a role in the nanoscale organization of IgM-BCR. In contrast, super-resolution dSTORM revealed no difference in the size or density of IgM-BCR nanoclusters in galectin-9-deficient B cells. This is consistent with a recent study demonstrating that glycan-based interactions do not mediate the nanoclustering of dendritic cell-specific intercellular adhesion molecule-3-grabbing non-integrin[52]. It may be that formation of BCR nanoclusters (or oligomers) is largely dependent on the direct interaction of BCR components, as suggested by Reth and colleagues[16,53]. However, we do find that addition of rGal9 induces significant changes in IgM-BCR organization. Inside the galectin-9 lattice, the number of IgM-BCR nanoclusters decreased while the radius of clusters, and the number of molecules in each cluster, increased. Thus, we propose that galectin-9 provides a second layer of organization to IgM-BCR by merging pre-existing nanoclusters.

But how does this observation square with our finding that rGal9 suppresses BCR signaling? Indeed, these observations are somewhat counter-intuitive given that BCR stimulation induces the coalescence of BCR nanoclusters to form larger-scale signaling microclusters[9]. So, how does the larger cluster of IgM induced by galectin-9 not activate B cells, but instead inhibit BCR signaling? We propose two potential mechanisms for galectin-9-mediated control of BCR signaling. First, we propose that galectin-9 induced merging of pre-existing nanoclusters decreases

These findings suggest that galectin-9 mediates close association between IgM and CD22, at least to some extent, in the steady state.

## Discussion

Here we identify galectin-9 as a negative regulator of B-cell activation. We found that BCR microcluster formation and signaling are enhanced in galectin-9-deficient B cells. Importantly,

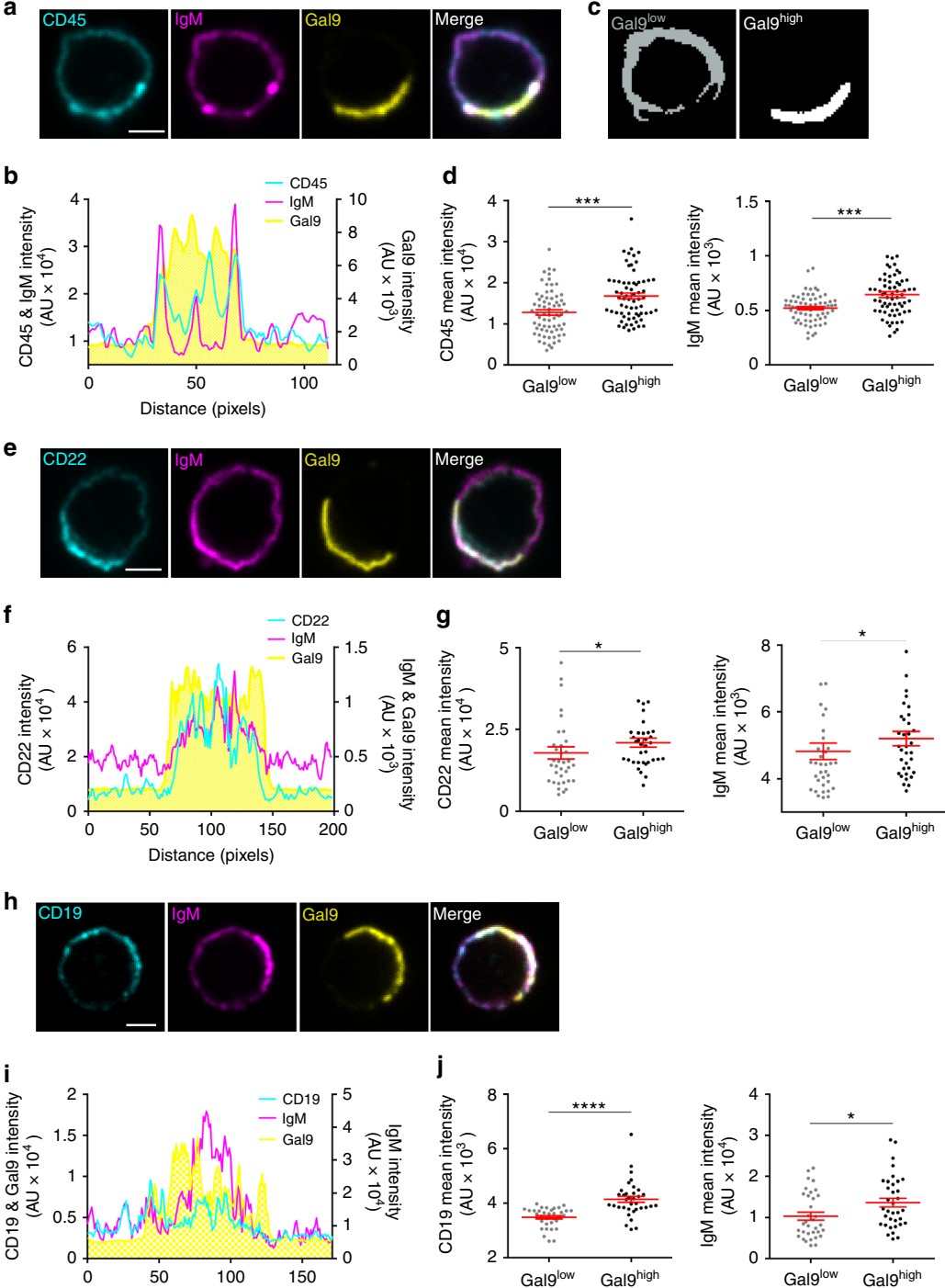

**Fig. 7** The galectin-9 lattice increases the molecular density of IgM-BCR and co-receptors. **a** Representative confocal images of primary WT B cells treated with 1 μM rGal9 and immunostained for CD45 (cyan), IgM (magenta), and galectin-9 (Gal9; yellow). **b** Fluorescence intensity profile of CD45, IgM, and Gal9 along the cell membrane. **c** Representative example of masking output of algorithm to detect regions of high galectin-9 (Gal9$^{high}$) and low galectin-9 (Gal9$^{low}$). **d** Mean fluorescence intensity of CD45 (left) and IgM (right) in Gal9$^{high}$ and Gal9$^{low}$ regions. **e** Representative confocal images of WT B cells treated with 1 μM rGal9 and immunostained for CD22 (cyan), IgM (magenta), and Gal9 (yellow). **f** Fluorescence intensity profile of CD22, IgM, and Gal9 along the cell membrane. **g** Mean fluorescence intensity of CD22 (left) and IgM (right) in Gal9$^{high}$ and Gal9$^{low}$ regions. **h** Representative confocal images of WT B cells treated with 1 μM rGal9 and immunostained for CD19, IgM, and Gal9. **i** Fluorescence intensity profile of CD19, IgM, and Gal9 along the cell membrane. **j** Mean fluorescence intensity of CD19 (left) and IgM (right) in Gal9$^{low}$ and Gal9$^{high}$ regions. Data representative of at least three independent experiments. Each dot represents 1 cell, at least 30 cells measured per condition per experiment. Mean ± SEM indicated by the red bar. Statistical significance assessed by Mann-Whitney; ****$p < 0.0001$ ***$p < 0.001$, *$p < 0.05$. Scale bar 2 μm

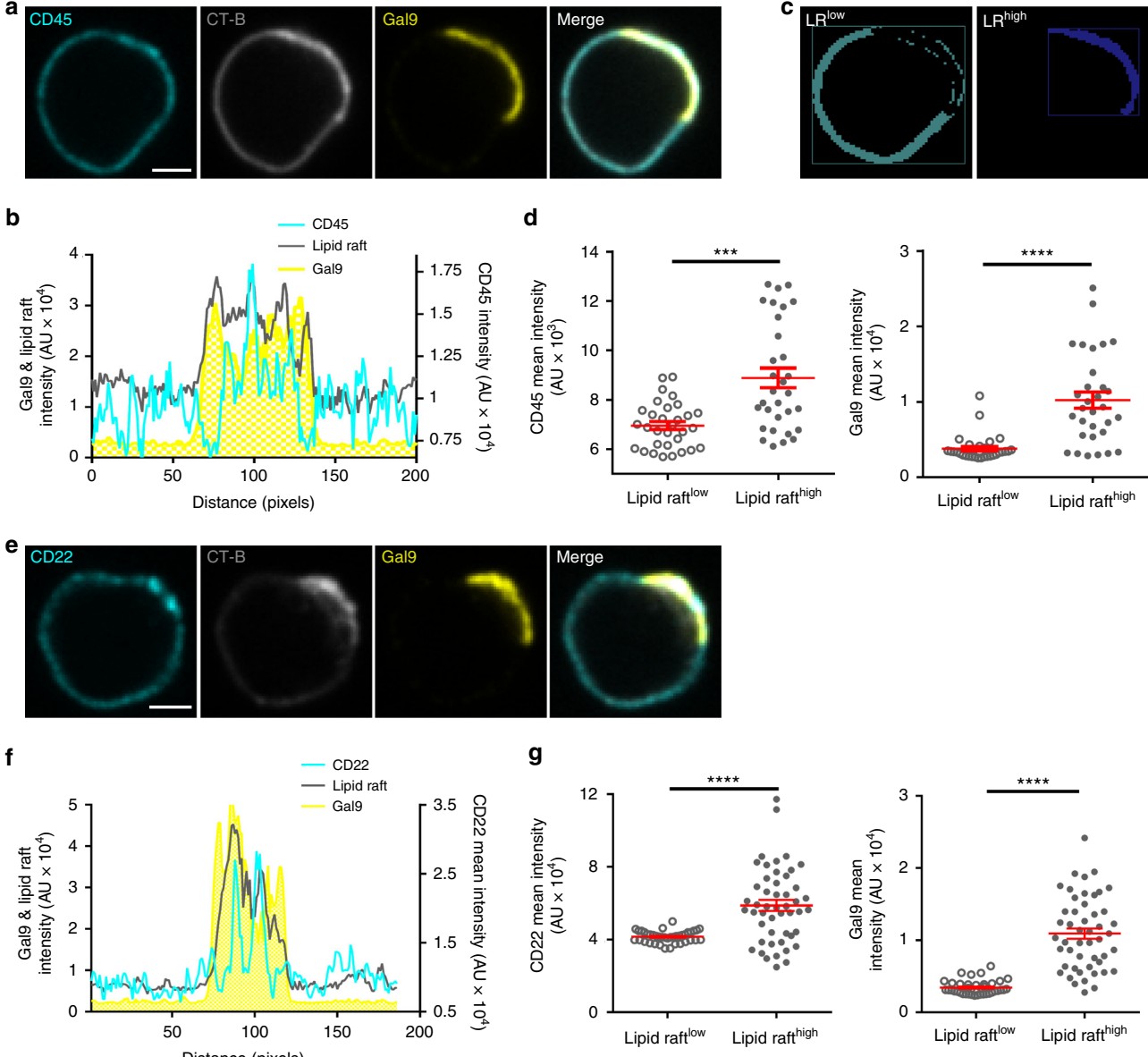

**Fig. 8** rGal9 induces coalescence of lipid raft domains containing CD22 and CD45. **a** Representative confocal images of primary WT B cells treated with 1 μM rGal9 and immunostained for CD45 (cyan) and galectin-9 (Gal9; yellow), and fluorescent cholera toxin (CT-B; gray) to label lipid rafts. **b** Fluorescence intensity profile of CD45, CT-B, and Gal9 along the cell membrane. **c** Representative example of masking output of algorithm to detect regions of high CT-B (lipid raft high; LR^high) and low CT-B (LR^low). **d** Mean fluorescence intensity of CD45 (left) and Gal9 (right) in LR^low and LR^high regions. **e** Representative confocal images of WT B cells treated with 1 μM rGal9 and immunostained for CD22 (cyan), Gal9 (yellow), and fluorescent CT-B (gray). **f** Fluorescence intensity profile of CD22, CT-B, and Gal9 along the cell membrane. **g** Mean fluorescence intensity of CD22 (left) and Gal9 (right) in LR^low and LR^high regions. Data are representative of at least three independent experiments. Each dot represents 1 cell, at least 30 cells measured per condition per experiment. Mean ± SEM indicated by the red bar. Statistical significance assessed by Mann-Whitney; ****$p < 0.0001$, ***$p < 0.001$. Scale bar 2 μm

the lateral diffusion of IgM-BCR and consequently suppresses BCR signaling. The mobility of IgM-BCR on the cell membrane can be described as Brownian motion restricted by the underlying actin cytoskeleton[12,54]. The theory of Brownian motion postulates that the larger the particle the slower the movement of the particle[55]. Thus, we would predict that larger clusters of IgM-BCR inside the galectin-9 lattice have a lower diffusion coefficient compared to smaller clusters outside the galectin-9 lattice, consistent with our data. This interpretation is supported by the finding that in CD45-deficient B cells, homotypic clustering of CD22 is enhanced, which correlates with decreased lateral diffusion[50]. In addition, IgD-BCR is more densely clustered than IgM-BCR[9] and this correlates with a 10-fold lower diffusion

coefficient compared to IgM-BCR[12]. Thus, we propose that galectin-9 restricts IgM-BCR mobility by organizing IgM-BCR into larger clusters. The mobility of BCR on the cell surface is correlated with BCR signaling; simply depolymerizing actin increases BCR mobility and triggers spontaneous signaling[12]. Consistent with this, treating B cells with lipopolysaccharide increases actin severing and BCR mobility, which lowers the threshold for B-cell activation[54]. Increased diffusion of the BCR (or BCR nanoclusters) may be important for BCRs to join newly forming microclusters. In agreement with this, treatment with rGal9 decreased BCR microcluster formation upon stimulation by membrane-bound antigen. Conversely, increased BCR diffusion would be associated with more microclusters, in line with our

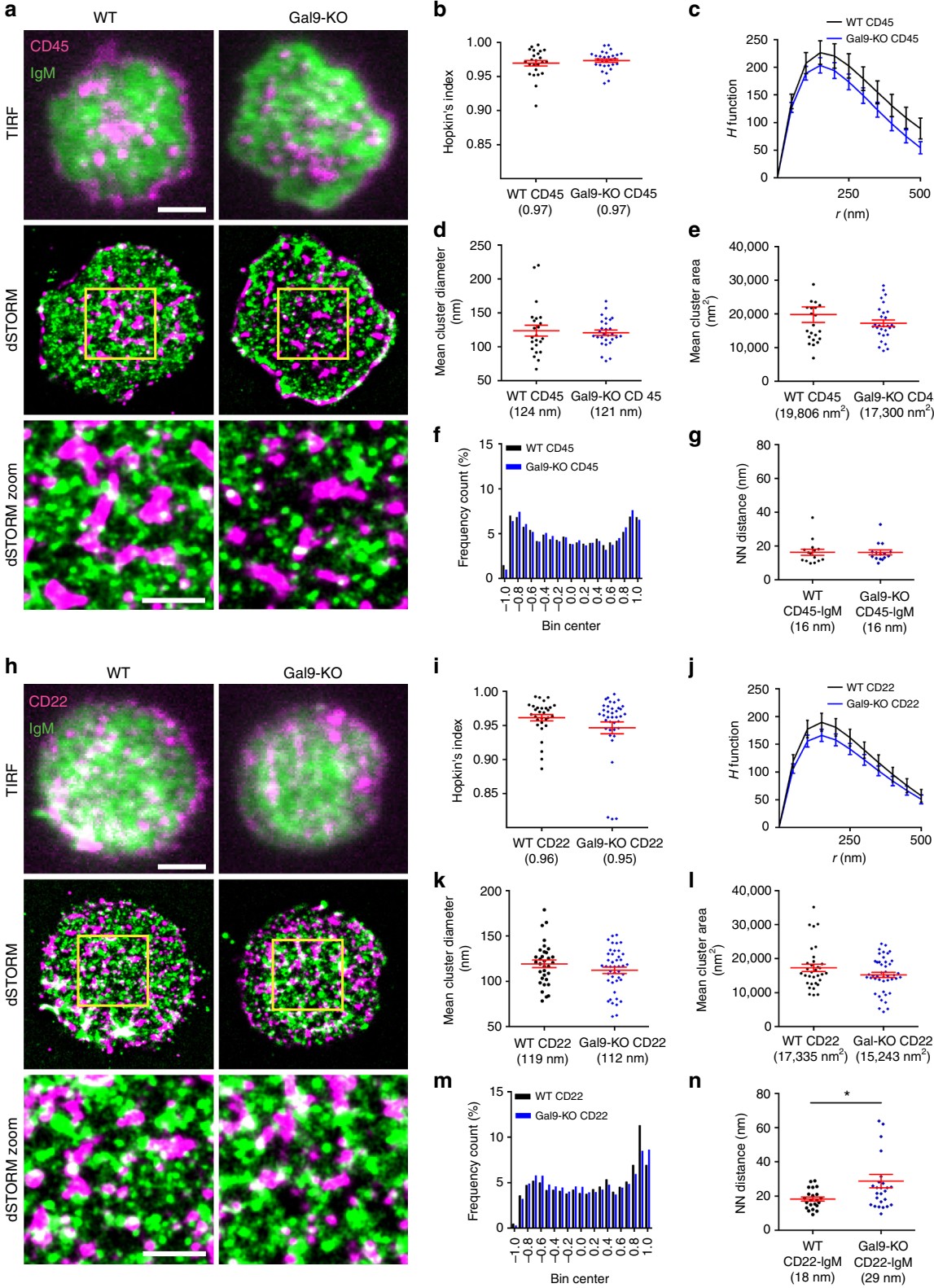

observation of increased BCR mobility in galectin-9-deficient B cells and enhanced B-cell activation.

Additionally, we propose that galectin-9 induced merging of BCR nanoclusters regulates the interaction between IgM and the co-receptors CD45 and CD22 to suppress BCR signaling. Our finding of CD45 enrichment in the galectin-9 lattice is consistent with our identification of CD45 as a ligand of galectin-9. The

N-terminal of CD45 has multiple *N*-glycosylation sites, providing potential binding sites for galectin family members. Indeed, in T cells, CD45 was reported to bind to galectin-1[56–58] and galectin-9[33]. In addition, galectin-3 has been reported to bind to CD45 on diffuse large B-cell lymphoma[59]. These findings suggest that CD45 may be a promiscuous ligand of the galectin family, although differential glycosylation of CD45 in different immune

**Fig. 9** Galectin-9 increases colocalization between CD22 and IgM in primary B cells. **a** Representative merged TIRF (top), dSTORM (middle), and dSTORM zoom (bottom) images showing surface CD45 (magenta) and IgM-BCR (green) on primary wild-type (WT) (left) and galectin-9 knockout (Gal9-KO) (right) B cells. dSTORM ROI (3 × 3 μm) is outlined in yellow (middle) and magnified in dSTORM zoom (bottom). **b–e** Quantification of at least 20 ROIs from WT and Gal9-KO B cells pooled from three independent experiments. **b** Hopkin's index showing randomness of CD45 organization (one point per ROI). **c** $H$ function derived from Ripley's $K$ showing degree of CD45 clustering. **d** Mean diameter of CD45 clusters (one point per ROI). **e** Mean area of CD45 clusters (one point per ROI). **f**, **g** Quantification of at least 15 ROIs from WT and Gal9-KO B cells pooled from three independent experiments. **f** Coordinate-based colocalization (CBC) histograms of the single-molecule distributions of colocalizations between CD45 and IgM. **g** Nearest-neighbor distance (NND) analysis of the data shown in **f**. Symbol represents the median NND of all paired single-molecule localizations from one ROI. **h** Representative merged TIRF and dSTORM images showing surface CD22 (magenta) and IgM-BCR (green) on primary WT (left) and Gal9-KO (right) B cells. **i–l** Quantification of at least 30 ROIs from WT and Gal9-KO B cells pooled from three independent experiments. **i** Mean Hopkin's index showing randomness of CD22 organization (one point per ROI). **j** $H$ function showing degree of CD22 clustering. **k** Mean diameter of CD22 clusters (one point per ROI). **l** Mean area of CD22 clusters (one point per ROI). **m**, **n** Quantification of at least 20 ROIs from WT and Gal9-KO B cells pooled from three independent experiments. **m** CBC histograms of the single-molecule distributions of colocalizations between CD22 and IgM. **n** NND analysis of the data shown in **m**. Colocalization between channels shown in white. Scale bars represent 2 and 1 μm (zoom). Mean ± SEM indicated by the red bar. Statistical significance assessed by Mann-Whitney, $*p < 0.05$

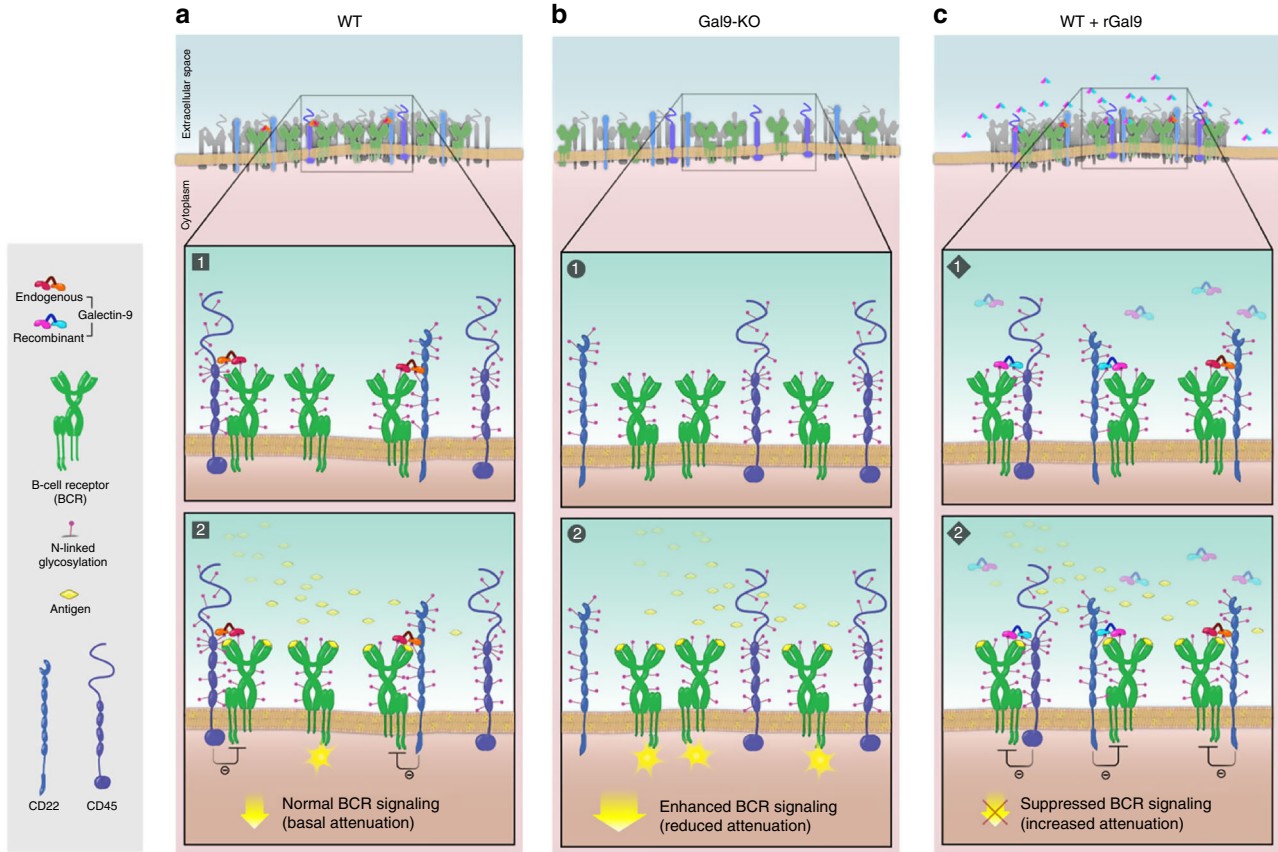

**Fig. 10** Schematic model of galectin-9 regulation of B-cell activation. **a** In resting primary naive WT B cells, galectin-9 facilitates interactions between BCRs and either the inhibitory proteins CD45 or CD22 through binding to N-linked glycans, providing a basal attenuation of B-cell signaling upon antigen stimulation. **b** BCR signaling is enhanced in Gal9-KO B cells due to loss of association of BCR with inhibitory co-receptors. **c** Treatment of WT B cells with rGal9 induces the association of IgM-BCR with CD45 and CD22 to suppress B-cell signaling

cell subsets may provide specificity for distinct galectin binding. Nonetheless, the enrichment of CD45 together with IgM inside the galectin-9 lattice is consistent with the inhibitory effect of galectin-9 on BCR signaling. Previous studies suggest that BCR activation is accompanied by the segregation of BCR and CD45 in order to promote Igα–Igβ phosphorylation[41,60], consistent with the finding that CD45 is excluded from BCR microclusters[2]. Our finding that addition of rGal9 induces co-patching of CD45 and IgM and concomitant abrogation of BCR signaling is consistent with this model.

We also observed the enrichment of CD22 inside the galectin-9 lattice. CD22 is a known negative regulator of BCR signaling, which recruits the phosphatase SHP-1[45], which dephosphorylates CD19[61] terminating the amplification of BCR signaling. In addition, SHP-1 dephosphorylates PLCγ2 and attenuates MAPK activation[42]. Thus, the localization of CD22 together with IgM–BCR upon rGal9 treatment is consistent with our observation of reduced phosphorylation of CD19 and ERK1/2 and likely contributes to the inhibitory effect of galectin-9 on B-cell activation. Moreover, in resting naive B cells, dual-color dSTORM

demonstrated that the colocalization of IgM and CD22 is reduced in galectin-9-deficient B cells, providing a plausible mechanism for enhanced BCR signaling in Gal9-KO B cells.

Interestingly, we found that the ligands for galectin-9 are not saturated in naive B cells and raises the possibility that BCR signaling could be modified under conditions that affect the secretion of galection-9. It is currently not known if B cells secrete galectin-9, which binds to the cell surface, or if other cells secrete galectin-9, which then binds to the surface of B cells. Indeed, galectin-9 is expressed in multiple cell types, including T cells, natural killer cells, and monocytes[62]. Alterations in galectin-9 secretion and its association with cell surface proteins would provide a mechanism to adjust the threshold of B-cell activation and thus tune B-cell responses. Moreover, recent reports indicate that galectin-9 is induced under inflammatory conditions in peripheral blood mononuclear cells and mesenchymal stromal cells[63], and elevated in the serum of patients in a variety of infections, including HIV, cytomegalovirus, and dengue infection[64–66]. How this affects the B-cell response in these infections is an important question to address.

## Methods

**Mice.** C57BL/6 (WT) mice were obtained from Charles River, $Lgals9^{-/-}$ (Gal9-KO) mice were obtained from Steven Beverley (Washington University) on behalf of The Scripps Research Institute, and $Lgals1^{-/-}$ (Gal1-KO) were obtained from The Jackson Laboratory. Mice were used at 2–6 months of age for all functional and biochemical experiments and were age- and sex-matched within each experiment. Mice were housed in specific pathogen-free animal facility at University of Toronto Scarborough, Toronto, Canada. All procedures were approved by the Local Animal Care Committee at the University of Toronto Scarborough.

**Cell lines and culturing.** A20 B cells and A20 expressing Hel-specific IgM (D1.3) were obtained from F. Batista (Ragon Institute), and Jurkat T cells were obtained from T. Watts (University of Toronto). Cell lines were not tested for mycoplasma contamination. Cells were maintained at 37 °C with 5% $CO_2$ in RPMI 1640 containing 10% heat-inactivated fetal bovine serum (FBS), 100 U/mL penicillin and streptomycin (all from Gibco), and 50 μM 2-mercaptoethanol (Amresco).

**Cell isolation.** Splenocytes were isolated from C57BL/6 WT, Gal9-KO, and Gal1-KO mice using 70 μm cell strainer and phosphate-buffered saline (PBS, pH 7.4, Life Technologies) to collect a single-cell suspension. B cells were purified using negative isolation kit (Miltenyi Biotec Inc. or Stem Cell technologies) according to the manufacturer's protocol.

**Immunohistochemistry.** Inguinal lymph nodes from C57BL/6 mice were isolated and immersed in Optimal Cutting Temperature compound (TissueTek) followed by submersion in 2-methylbutane cooled in liquid nitrogen. Frozen tissue blocks were cryosectioned into 12 μm slices and adhered to Superfrost Plus microscope slides (VWR), fixed in 4% paraformaldehyde (PFA), and then air-dried. Tissue slices were rehydrated with PBS for 15 min, permeabilized with 0.5% Tween-20 in PBS for 5 min, then blocked with 5% goat serum in PBS for 30 min. Sections were incubated with CD169-FITC (clone 3D6.112, Biolegend), B220-allophycocyanin (APC) (clone RA3-6B2, Invitrogen), and goat anti-mouse galectin-9 (R&D Systems, Cat. No. AF3535) at 1:200 v/v in blocking buffer for 1 h at room temperature. Sections were washed five times with 0.5% Tween-20 in PBS, and then incubated with secondary antibody Cy3-conjugated bovine anti-goat (Jackson Immunoresearch, Cat. No. 805-165-180) at 1:1000 v/v for 1 h at room temperature. Slides were washed five times and then mounted in Fluoro-Gel with DABCO™ Antifading Mounting Medium. Slides were imaged by laser scanning confocal microscopy with Zeiss LSM510 Meta using a ×100/numerical aperture (NA) 1.40 oil-immersion objective.

**Surface staining of galectin-9 for flow cytometry and imaging.** C57BL/6, Gal9-KO, or rGal9-treated B cells were first blocked with purified rat anti-mouse CD16/32 (Fc-block, 1:250 v/v, clone FCR-4G8, Invitrogen) in fluorescence-activated cell sorting (FACS) buffer (1% bovine serum albumin (BSA), 4 mM EDTA, and 0.01% $NaN_3$ in PBS) for 30 min at 4 °C. Cells were washed three times with FACS buffer and then immunostained with phycoerythrin-conjugated rat anti-mouse galectin-9 antibody (clone 108A2, Biolegend) at 1:200 v/v, or stained with goat anti-galectin-9 antibody (R&D Systems, Cat. No. AF3535) at 1:100 v/v in FACS buffer for 30 min at 4 °C, followed by Cy3-conjugated AffiniPure bovine anti-goat antibody (Jackson ImmunoResearch Laboratories, Cat. No. 805-165-180) at 1:1000 v/v in FACS buffer for 30 min at 4 °C. Cells were then stained with the following antibodies: APC-conjugated anti-CD19 (clone ID3, Biolegend) at 1:200 v/v; Alexa Fluor 488 or

Alexa Fluor 644 AffiniPure Fab fragment rat anti-mouse IgM, μ-chain-specific (Jackson ImmunoResearch Laboratories, Cat. No 115-547-020 or Cat. No 115-607-020) at 1:200 v/v in FACS buffer for 30 min at 4 °C. Cells were washed and resuspended in FACS buffer for flow analysis or imaged by spinning disc confocal microscopy. Cells were analyzed by flow cytometry (BD Fortessa) and plotted using Flowjo software (TreeStar). Cells were imaged by spinning disc confocal microscopy (Quorum Technologies) consisting of an inverted fluorescence microscope (DMI6000B; Leica) equipped with a ×63/1.4 NA oil-immersion objective and an electron-multiplying charge-coupled device (EMCCD) camera (ImageEM; Hamamatsu). Fluorescence intensity images of galectin-9 were pseudo-colored using the fire lookup table in ImageJ. Quantification of number of puncta for all z-sections was performed in ImageJ using a threshold of 2× background intensity followed by "Analyze Particles" algorithm with the following parameters: size 5–100; circularity 0.00–1.00.

**Planar lipid bilayer.** Artificial planar lipid bilayers were prepared by liposome spreading[67] in FCS2 chambers (Bioptechs). 1, 2-dioleoyl-sn-glycero-3-phosphocoline liposomes (Avanti Polar Lipids, Inc.) were mixed with $2.5 \times 10^{-4}$% biotinylated (N-Biotinyl Cap PE; Avanti Polar Lipids, Inc) liposomes. Alexa Fluor 633-streptavidin (1:1000 v/v Life Technologies, Cat. No S21375) was used to tether monobiotinylated anti-mouse kappa light chain (clone HB58, purified in house from hybridoma) as surrogate antigen. Cells were assayed in chamber buffer (0.5% FBS, 2 mM $MgCl_2$, 0.5 mM $CaCl_2$, and 1 g/L D-glucose in PBS). Freshly isolated splenocytes from WT and Gal9-KO mice were allowed to spread and form clusters for 90 s, and then fixed with 4% PFA for 15 min at 37 °C. Cells were imaged using a spinning disc confocal microscope as described above. Images were analyzed using Volocity (Perkin Elmer) to quantify area of cell-bilayer contact, total antigen intensity, and mean antigen intensity, and plotted using Prism6 (GraphPad). Antigen was pseudo-colored using the fire lookup table provided in ImageJ (National Institutes of Health).

**Pull-down assay and mass spectrometry.** The protocol was adapted from ref.[33]. In all, $10^7$–$5 \times 10^7$ purified B cells from Gal9-KO mice were lysed in NP40 lysis buffer (50 mM HEPES (pH 8.0), 150 mM NaCl, 5 mM dithiothreitol (DTT), 5 mM EDTA, 0.1 % Nonidet-P40, and Roche cOmplete mini EDTA-free protease inhibitor) with three cycles of freeze/thaw using liquid nitrogen. Lysates were cleared by centrifugation at 21 000 × g for 10 min at 4 °C. The cell lysate was incubated with anti-FLAG beads (Sigma-Aldrich) conjugated with 10 μg FLAG-tagged rGal9 (kindly provided by M. Ostrowski, University of Toronto) for 3 h at 4 °C. Beads were subjected to magnetic separation and washing in lysis buffer. Bound proteins were first eluted by 200 mM lactose solution (in distilled water), followed by denaturing elution (125 mM $NH_4OH$). Denatured eluents were evaporated to dryness to remove $NH_4OH$ using a vacuum centrifuge, followed by resuspension in 50 mM ammonium bicarbonate (pH 8.0). Both eluents were then subjected to in-solution tryptic digestion followed by peptide desalting and concentration using OMIX C18 tips (Agilent) according to the manufacturer's instructions. Eluted peptides were dried by vacuum centrifugation and stored at −20 °C prior to analysis. Peptide separations were performed with an increasing acetonitrile gradient (5–40% acetonitrile and 0.1% formic acid) over 90 min using an EASY nLC-1000 HPLC equipped with a 10.5 cm C18 column coupled online to an LTQ XL mass spectrometer (Thermo). MS1 scans were acquired between 400 and 1800 m/z, and up to six of the most intense parent ions were selected for collision-induced dissociation with an isolation width of 2 Da and normalized collision energy of 35 eV. Parent ions selected for analysis more than three times in 30 s were dynamically excluded from analysis for 45 s. Raw spectra were converted to mzXML format using msConvert (PMID: 23051804) and searched against the mouse and Escherichia coli proteomes as well as the sequence of Lgals9 and a database of commonly observed contaminants using X! Tandem (PMID: 14976030). Parent ion tolerances were set to +3 and −2 Da and fragment ion mass tolerance was set to 0.4 Da. Oxidation of methionine and tryptophan, as well as deamidation of asparagine and glutamine was permitted as potential modifications during refinement. Results were further compared using ProhitsVM[68].

**Pull-down assay for western blotting.** A total of 1–3 × 10^7 purified B cells from Gal9-KO mice were lysed in RIPA buffer (10 mM Tris-HCl (pH 8.0), 1 mM EDTA, 1% Triton X-100, 0.1% sodium deoxycholate, 0.1% SDS, 140 mM NaCl, and 1 mM phenylmethylsulfonyl fluoride (PMSF)). Lysates were cleared by centrifugation at 21 000 × g for 10 min at 4 °C. Cell lysates were incubated with anti-FLAG beads (Sigma-Aldrich) conjugated with 10 μg FLAG-tagged rGal9 for 3 h at 4 °C in the presence or absence of 50 mM D-lactose. Beads were subjected to magnetic separation and washed in RIPA buffer. Bound proteins were eluted using 200 mM D-lactose in PBS. Protein was mixed with equal volume of 2× Laemmli sample buffer, heated to 95 °C for 10 min, and separated by SDS-polyacrylamide gel electrophoresis (SDS-PAGE). Proteins were transferred to polyvinylidene fluoride (PVDF) membrane and blocked with 5% non-fat milk in TBST (20 nM Tris (pH 7.5), 150 mM NaCl, and 0.1% Tween-20) for 30 min at room temperature with gentle shaking. Primary antibodies rat anti-mouse CD45RA (clone 30-F11; BioLegend) and goat anti-mouse IgM (Jackson Immunoresearch, Cat. No. 115-005-020) were incubated at 1:1000 overnight at 4 °C with gentle shaking. Secondary

antibodies horseradish peroxidase (HRP)-conjugated donkey anti-rat and donkey anti-goat (Abcam, Cat. No ab102182 and ab97110) were incubated at 1:10 000 in 5% non-fat milk in TBST for 1 h at room temperature with gentle shaking. Membranes were washed with TBST and developed with chemiluminescent substrate (West Pico, Thermo Fisher), and imaged by ChemiDoc (Bio-Rad).

**Far-western blotting**. In all, $1–5 \times 10^7$ purified primary murine B cells, D1.3 B cells, A20 B cells, or Jurkat T cells were lysed in RIPA buffer (10 mM Tris-HCl pH 8.0; 1 mM EDTA, 1% Triton X-100, 0.1% sodium deoxycholate, 0.1% SDS, 140 mM NaCl, and 1 mM PMSF). Lysates were cleared by centrifugation at $21 000 \times g$ for 10 min at 4 °C. A unit of 25 μg of whole-cell lysate was resolved by 10% SDS-PAGE and transferred to PVDF membrane. Proteins were re-natured in decreasing concentrations of guanidine-HCl (6–0 M) according to ref.[34]. Membranes were then blocked with 5% non-fat milk in TBST for 30 min at room temperature with gentle shaking. A unit of 5 μg/mL FLAG-tagged-recombinant galectin-9 in 2.5% skim milk in TBST was used as a bait protein, blots were incubated with bait protein overnight at 4 °C with gentle shaking. Blots were then washed and incubated with anti-FLAG (clone M2, Sigma-Aldrich) at 1:1000 in 5% skim milk in TBST for 1 h at room temperature. Blots were washed and incubated with HRP-conjugated anti-mouse antibody (Jackson Immunoresearch, Cat. No. 115-035-003) at 1:10 000 in 5% non-fat milk in TBST for 1 h at room temperature with gentle shaking. Membranes developed with chemiluminescent substrate (West Pico, Thermo Fisher) and imaged by ChemiDoc (Bio-Rad).

**rGal9 and Fab fragment labeling for single-particle tracking**. Mouse rGal9 (R&D Systems) was reconstituted at 0.2 mg/mL in 0.1 M NaHCO3 and incubated with 0.2 mg/mL Alexa Fluor® 555 NHS Ester for 1 h at room temperature with gentle mixing. Following labeling, the mixture was dialyzed against 20 mM MOPS, 500 mM sodium chloride, 0.5 mM EDTA, and 1 mM DTT using a 10 000 molecular weight cutoff Slide-A-Lyzer® Dialysis Cassette (Thermo Scientific). A volume of 20 μl of 1 M NaHCO3 was added to 200 μl of 1 mg/mL goat anti-mouse IgM, μ-chain-specific (Jackson ImmunoResearch, Cat. No. 115-007-020) and incubated with 40 μg/mL Attotec® 633 NHS Ester for 1 h at room temperature with gentle mixing. Following labeling, the mixture was dialyzed against PBS.

**Glass coverslip coating for single-particle tracking**. Glass coverslips were cleaned in chromic acid (Fisher) for 20 min followed by rinsing with water and acetone. Coverslips were air-dried and then incubated with 1 μg/mL anti-MHC class II (clone M5/114, Sunnybrook Research Institute) for 2 h at room temperature and then washed with PBS.

**Cell labeling for single-particle tracking**. Primary murine B cells from WT and Gal9-KO mice were labeled with 4 ng/mL Attotec® 633-labeled goat anti-mouse IgM Fab fragment (Jackson ImmunoResearch, Cat. No. 115-007-020) in 0.5% FBS in PBS for 15 min at 4 °C. For rGal9 treatment, $5 \times 10^6$ B cells from Gal9-KO mice were incubated with 0.5 μM labeled rGal9 mixed with 0.5 μM non-labeled rGal9 in complete media for 30 min at 37 °C. Labeled cells were washed twice with PBS and resuspended in chamber buffer. Labeled cells were stored on ice prior to imaging. Just before imaging, cells were incubated at 37 °C for 5 min.

**Instrument for single-particle tracking and dSTORM**. Single-molecule fluorescence microscopy was performed with a TIRF microscope (Quorum Technologies) based on an inverted microscope (DMI6000B; Leica), HCX PL APO ×100/1.47 oil-immersion objective, and Evolve Delta EMCCD camera (Photometrics). Images were acquired continuously at 20 frames/s for 10 s with an EM gain of 200 and the exposure time of 50 ms.

**Single-particle tracking**. We used the well-established single-particle tracking algorithm by Crocker and Grier[69] implemented in Matlab (The MathWorks) by Daniel Blair and Eric Dufresne (http://physics.georgetown.edu/matlab/). Particle positions were determined by measuring the the centroid position with sub-pixel accuracy based on the local maxima and then linking particle positions. Track lengths < 10 were discarded in order to reduce statistical noise. Diffusion coefficients for individual trajectories were determined form the first three data points of the mean squared displacement curves of each trajectory.

**Sample preparation for dSTORM**. Primary B cells from WT and Gal9-KO mice were stained with Alexa Fluor 647-conjugated Fab fragment goat anti-mouse IgM, μ-chain-specific (Jackson ImmunoResearch, Cat. No. 115-607-020) at 1.5 μg/mL in PBS containing 2% FBS for 15 min at 4 °C, then washed twice with PBS. For dual-color dSTORM, primary WT and Gal9-KO B cells were stained with Alexa Fluor 488-conjugated Fab fragment goat anti-mouse IgM, μ-chain-specific (Jackson ImmunoResearch, Cat. No 115-547-020) at 2.8 μg/mL and either Alexa Fluor 647 anti-mouse CD22 (OX-97, Biolegend) at 2.5 μg/mL or Alexa Fluor 647 anti-mouse/human B220 (RA3-6B2, Biolegend) at 2.5 μg/mL all in PBS containing 2% FBS for 20 min at 4 °C, then washed twice with PBS. Cells were allowed to spread on anti-MHC-class II-coated coverslips for 10 min at 37 °C. Coverslips were washed gently with PBS to remove unbound cells. Cells were fixed with fixation solution (4% PFA

and 0.2% glutaraldehyde in PBS) for 40 min at room temperature. The coverslips were washed three times with PBS. Before single-color dSTORM imaging, samples were incubated in PBS containing 0.1 M mercaptoethylamine, 0.5 mg/mL glucose oxidase, 40 μg/mL catalase, and 10% glucose. For dual-color dSTORM, samples were incubated in PBS containing 0.1 M β-mercaptoethylamine hydrochloride (Sigma-Aldrich), 3% (v/v) OxyFlour™ (Oxyrase Inc., Mansfield, Ohio, USA), 20% (v/v) of sodium DL-lactate solution (L1375, Sigma-Aldrich) adjusted to pH ~ 8.3. Fiducial markers (100 nm TetraSpeck Fluorescent Microspheres, Invitrogen) were added to buffer and allowed to settle for 5 min prior to imaging.

**dSTORM acquisition and image reconstruction**. dSTORM images were acquired on a TIRF microscope as described above. For Alexa Fluor® 647, photoconversion was achieved with 633-nm laser (intensity ranged from 80 to 100 mW/cm²) illumination and conversion from the dark state with 488-nm laser illumination (intensity range from 5 to 20 mW/cm²). Ten thousand images were acquired at a frame rate of 33 frames/s. Dual-color dSTORM images were acquired sequentially on a TIRF microscope, first imaging Alexa 647 followed by Alexa 488 to prevent photobleaching of Alexa 647. For Alexa Fluor® 647, photoconversion was achieved with 633-nm laser (intensity ranged from 80 to 100 mW/cm²) illumination and conversion from the dark state with 402-nm laser illumination (intensity range from 5 to 20 mW/cm²). For Alexa Fluor® 488, photoconversion was achieved with 488-nm laser (intensity ranged from 80 to 100 mW/cm²). Eight thousand images were acquired at a frame rate of 30 frames/s. Fiducial markers, which were visible in both the 488- and 647-nm channels, were used to align the two channels. The images of the beads in both channels were used to calculate a polynomial transformation function that mapped the 488-nm channel onto the 647-nm channel, using the MultiStackReg plugin of ImageJ. The transformation matrix was applied to each frame of the 488-nm channel stack. Reconstructed images were acquired using ThunderSTORM plugin for ImageJ[70] according to the parameters shown in Supplementary Fig. 8A. Briefly, camera setup was as follows: pixel size (101.5 nm); photoelectron per A/D count 2.4; base level [A/D count] 414; and an EM gain of 50 (dSTORM) or 200 (dual-color dSTORM). Image filtering was applied to remove camera noise and enhance photoswitching events using a wavelet filter (B-Spline) with B-Spline order of 3 and B-Spline scale of 2.0. Approximate localization of molecules was detected by local maximum method with a peak intensity threshold of std (Wave.F1) and a connectivity of 8-neighborhood. Sub-pixel localization of molecules was identified by fitting point spread function to an integrated Gaussian using weighted least squares method with a fitting radius of 3 pixels. Single molecules may not be adequately resolved in spatially dense organizations and thereby multiple activated molecules are detected as a single blob. Multiple emitter fitting analysis was used to estimate the number of molecules detected as a single blob with maximum five molecules per fitting region. To improve the multi-emitter fitting algorithm, molecules are fitted assuming the same intensity. Super-resolution images were rendered with a pixel size of 20 nm.

**dSTORM post-processing, analysis, and quantification**. Reconstructed dSTORM images were post-processed with drift correction using the built-in method in the ThunderSTORM plugin. The workflow for post-processing steps is shown in Supplementary Fig. 8B and consisted of the following steps: (1) remove duplicates, in which repeated localizations of single molecules in one frame, which may occur when using multiple emitter, were removed based on uncertainty radius of localization; (2) filter, in which localizations with an uncertainty >20 nm were eliminated; (3) density filter, in which isolated localizations were removed based on the parameter of at least two neighbors are required in 50 nm radius for a localization to be accepted; (4) drift correction, in which image drift was corrected using fiducial markers; and (5) merging, in which molecules that appeared within 20 nm in multiple frames were merged together. The number of localizations across cells and conditions was similar (Supplementary Fig. 8C). Performance evaluation of dSTORM analysis and post-processing parameters was performed (Supplementary Methods; Supplementary Fig. 9). To reduce processing time, 4000 frames were processed for cluster analysis. For each reconstructed image of WT and Gal9-KO B cells, a $3 \times 3$ μm region was manually selected but excluded from the cell boundary. For each reconstructed image of rGal9-treated Gal9-KO B cells, a $3 \times 3$ μm region that colocalized with Gal9, was manually selected for analysis. The Hopkins index and Ripley's $H$ function analysis were performed by SuperCluster, an analysis tool kindly provided by the University of New Mexico's Spatio Temporal Modeling Center (http://stmc.unm.edu/). Cluster analysis was performed by a Bayesian, a model-based approach[38]. In brief, uncertainty, $x$ and $y$ coordinates of each localization in post-processed reconstructed images were exported. The selected $3 \times 3$ μm regions were analyzed by the published R code of Bayesian cluster analysis[38] with an alpha value of 20, pbackground of 0.5, rseq of (5, 200, 10), and thseq of (0, 50, 5). The analyzed data were post-processed to extract data about percentage of molecules localizing in clusters, cluster radius, number of clusters, and number of molecules per clusters.

**CBC analysis**. Colocalization of dual dSTORM data was conducted using CBC analysis[49] using ThunderSTORM. Briefly, this method uses coordinate information of each molecule instead of an intensity-based approach, which would depend on the reconstruction and post-processing parameters chosen. Furthermore, this

method takes into account the spatial distribution of each set of localizations to prevent false positive colocalization values that result when one of the molecular species is randomly organized. First, the spatial distribution function of the neighboring localizations from the same species in each channel are calculated. Then, from the individual distribution functions, a correlation coefficient is calculated and weighted by distance to the nearest neighbor of the localization's respective species. As a result, each single molecule of each species is attributed an individual colocalization value, which provides information on the molecule's local environment. CBC algorithm was applied to $x$–$y$ coordinate lists of localizations in the $3 \times 3$ µm region of 647 and 488 channels. A search radius of 70 nm, based on the radius IgM nanoclusters, was used to calculate degree of colocalization values varying from $-1$ (perfectly excluded) to $+1$ (perfectly colocalized).

**Cell stimulation for western blot**. Twenty-four-well plates were coated with 5 µg/mL AffiniPure F(ab′)2 fragment goat anti-mouse IgM, µ-chain-specific (Jackson ImmunoResearch Laboratories, Cat. No 115-006-020) in PBS overnight at 4 °C. Primary naive splenic B cells from WT and Gal9-KO mice were stimulated for the indicated time, then lysed with Laemmli buffer containing 100 µM DTT followed by SDS-PAGE. For rGal9 treatment, Gal9-KO or WT B cells were pre-treated with 0.1 or 1 µM, respectively, with recombinant mouse galectin-9 (rGal9, E. coli-derived, R&D Systems) in RPMI containing 1% FBS (both Gibco) and incubated for 30 min at 37 °C. Cells were then washed once with RPMI and then stimulated as described above. Proteins were transferred to PVDF and blocked with 5% non-fat skimmed milk or 5% BSA in TBST for 1 h at room temperature or overnight (for 4G10) at 4 °C with shaking. Membranes were immunoblotted with the following antibodies overnight or for 3 h (for 4G10) at 4 °C: mouse anti-β-tubulin (Sigma, 1:5000 in 1% BSA/TBST); rabbit anti-phospho-p44/42 MAPK (ERK1/2 Cell Signaling, Cat. No. 9101, 1:1000 in 1% BSA/TBST); rabbit anti-ERK p44/42 MAPK (ERK1/2 Cell Signaling, Cat. No. 9102); 4G10 (Sunnybrook Research Institute, 1:1000 in 1% BSA/1% NaN₃/TBST); phospho-CD19 (Tyr531 Cell Signaling, Cat. No. 3571, 1:1000 in 1% BSA/1% NaN₃/TBST); and phospho-Akt (Tyr308 Cell Signaling, Cat. No. 4056, 1:1000 in 1% BSA/1% NaN₃/TBST). HRP-conjugated secondary antibodies were used at 1:10 000 v/v in 5% non-fat skimmed milk in TBST or 1% BSA in TBST and incubated for 2 h with shaking at room temperature. Membranes were incubated with SuperSignal (Thermo Fisher Scientific) chemiluminescent substrate and then imaged by ChemiDoc (Bio-Rad). Densitometric analysis of western blots was performed using ImageJ (NIH). The amount of phosphorylated protein was normalized to loading control and then expressed as a fold change relative to the 0 min time point. Full blots for all cropped blots presented in the figures are shown in Supplementary Fig. 10.

**Calcium signaling**. Intracellular $Ca^{2+}$ flux was measured by flow cytometry. Freshly isolated splenocytes ($5 \times 10^6$ cells/sample) were labeled with 1 µM Fluo-4, AM (Thermo Fisher) in 10% FBS/Hank's balanced salt solution at 37 °C for 15 min. Cells were incubated with anti-CD45R (B220) (clone RA3-6B2; Biolegend) for 5 min at 4 °C at a dilution of 1:200. Cells were washed 2× with PBS and resuspended in RPMI. After collecting a baseline reading for 30 s, 20 µg/mL anti-mouse kappa immunoglobulin light chain (clone HB58) or 1 µM rGal9 was added to the facs tube, and change in fluorescence intensity was recorded on a Fortessa cytometer (BD Biosciences) and plotted using Flowjo software (TreeStar). Fold change was calculated by dividing the fluorescence intensity at each time point by baseline intensity.

**rGal9 treatment and confocal imaging**. Eight-well Lab-Tek chambers were coated with 1 µg/mL anti-mouse MHC II (M5/114, Sunnybrook Research Institute) antibody in PBS for 2 h at room temperature. In all, $5 \times 10^6$ primary murine B cells from WT mice were treated with 1 µM recombinant galectin-9 in 2% FBS in RPMI for 30 min at 37 °C. Cells were washed 1× with PBS. Cells were resuspended in PBS and allowed to adhere to chambers for 15 min. Cells were fixed in 2% PFA for 10 min and then washed 2× in FACS buffer (PBS, 1% BSA, and 0.1% NaN₃). Cells were incubated with 2 µg/mL purified rat anti-mouse CD16/32 (clone 2.4G2, BDPharmingen) in PBS for 15 min at 4 °C. Cells were incubated with 1 µg/mL goat anti-mouse Gal9 (R&D systems, Cat. No. AF3535) in FACS buffer for 1 h at 4 °C. Cells were washed 3× with FACS buffer. If cells were stained to visualize CD16/32, cells were incubated with 1 µg/mL Alexa-Fluor594 Donkey Anti-Rat IgG (Invitrogen Cat. No A-21209) for 1 h at 4 °C. Cells were washed 2× with PBS. Cells were incubated with 1 µg/mL Cy3-conjugated bovine anti-goat IgG (Jackson ImmunoResearch, Cat. No. 805-165-180) or Alexa Fluor 488 donkey anti-goat IgG secondary (Jackson ImmunoResearch, Cat. No. 705-545-147) for 1 h at 4 °C. Cells were washed 3× with PBS. Cells were stained with fluorochrome-conjugated antibodies specific for CD45/B220 (RA3-6B2, BD Biosciences), CD22 (OX-97, Biolegend), CD19 (1D3, Biolegend), IgD (11-26c.2a, Biolegend) at 1:100 v/v or IgM (Jackson ImmunoResearch) at 1:1000 v/v for 1 h at 4 °C. Cells were washed 3× with PBS. Cells were mounted in Fluoro-Gel with DABCO™ (Electron Microscopy). Chambers were imaged by spinning disk confocal microscopy (Quorum Technologies) consisting of an inverted fluorescence microscope (DMI6000B; Leica) equipped with a ×63/1.4 NA oil-immersion objective and an EMCCD camera (Image EM; Hamamatsu). Images were acquired using Metamorph software (Molecular Devices). Images were analyzed using Volocity software (Perkin Elmer). The fluorescence signal of CD45, CD22, CD19, IgM, IgD, or FcγRIIb was used to define a mask delineating the membrane region. Gal9$^{high}$ regions were determined by the fluorescence signal of Gal9. Gal9$^{low}$ regions were determined by subtracting the Gal9$^{high}$ regions from the membrane region. The mean fluorescence intensity of CD45, CD22, CD19, IgD, FcγRIIb, and IgM was calculated in Volocity.

**Lipid raft staining**. Primary murine WT B cells and 1 µM rGal9-treated B cells were labeled with 1 µg/mL Alexa Fluor 555-conjugated Cholera toxin subunit B (Vybrant™ Alexa Fluor™ 555 Lipid Raft Labeling Kit, Thermo Fisher Scientific) for 15 min on ice. Cells were washed 2× with PBS for 5 min at 4 °C. Cells were then allowed to adhere to anti-MHC II-coated Mattek chambers for 10 min. Cells were fixed with 2% PFA for 15 min and washed 3× with PBS. Cells were then stained for specific proteins as described above.

**B-cell apoptosis**. Freshly isolated naive B cells from WT and Gal9-KO mice were treated with 1 µM recombinant mouse galectin-9 (rGal9, E. coli-derived, R&D Systems) for 30 min at 37 °C, followed by staining with Fluorescein isothiocyanate (FITC)-conjugated Annexin V and propidium iodide according to the manufacturer's protocol (Annexin V Apoptosis Detection Kit, eBioscience), and analyzed by flow cytometry (BD Fortessa). Data were analyzed using FlowJo software.

**Statistical analysis**. Statistical analysis was performed using GraphPad Prism. The distribution of data was tested using D'Agostino-Pearson omnibus normality test. Comparisons between two groups were performed using Mann-Whitney for data with non-normal distribution. Comparisons between multiple groups were performed by ordinary one-way analysis of variance (ANOVA) for data with normal distribution and Kruskal-Wallis test for data with non-normal distribution. For post hoc analysis, Tukey's multiple comparison was used for normally distributed data and Dunn's multiple comparisons test was used for non-normally distributed data. Western blot data were analyzed by two-way ANOVA followed by Sidak's multiple comparisons test. Sample sizes were chosen based on prior literature using similar methods and reporting moderate effect size. Estimate of variance is reported as standard error of the mean for each data set.

**Code availability**. ThunderStorm plugin for ImageJ is available at http://zitmen. github.io/thunderstorm/. SuperCluster is available from the University of New Mexico's Spatio Temporal Modeling Center (http://stmc.unm.edu/). Bayesian cluster analysis code is available in supplementary information in ref. [38]. Single-particle tracking algorithms implemented in Matlab (The MathWorks) is available at http://physics.georgetown.edu/matlab/.

**Data availability**. Lgals9−/− mice require an MTA from The Scripps Research Institute (TSRI). The data that support the findings of this study are available from the corresponding author upon reasonable request.

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

## Acknowledgements

We thank Kiera Clayton and Mario Ostrowski for human rGal9[33]; Jose Grespan Guerios for help with dSTORM acquisition; Enayat Mohammad Gul for help with pull-down assays; Wen Zheng for help with cryosectioning and immunostaining of lymph nodes; and Jerry Gu for preparation of schematic model. This work was supported by funding from the Canadian Institutes of Health Research (CIHR; MOP-136808), Natural Sciences and Engineering Council of Canada (NSERC; RGPIN 418756-12), and Canada Research Chair (CRC; 905–231134) to B.T.; King Abdullah Scholarship to N.A.; NSERC graduate student fellowship to L.W.; and NSERC Undergraduate Student Research Award (USRA) to H.F. and W.Z.

## Author contributions

A.C., N.A. and B.T. designed the research study. A.C., N.A., F.H.M.B., L.W., L.K.S., A.T.Q., Z.H., H.F. and B.T. conducted the experiments and data analysis. M.S. and D.M. O. provided input and assistance on dSTORM analysis. A.T.Q. and A.S. performed mass spectrometry analysis. A.C., N.A., F.H.M.B., L.W., L.K.S. and B.T. wrote the manuscript.

## Additional information

**Competing interests:** The authors declare no competing interests.

