## [Peer Review File · Nature Communications]

Reviewers' comments:

Reviewer #1 (Remarks to the Author):

In this manuscript the authors explore the effect of the binding of the evolutionarily conserved lectin, galectin-9 (gal-9) on the spatial organization of IgM-containing B cell receptors (BCRs) and their signaling function. The role of galectin in modulating B cell responses is an area of genuine interest as glycan-binding proteins have been established to play a central role in the regulation of the immune system. As the authors point out galectins have been linked to immune regulation in lymphocytes in particular in T cells through their binding to CD45. Consequently, information on the mechanisms by which galectins function in B cell activation are of interest. Unfortunately, even though these expert authors use cutting edge biochemical and imaging techniques to address the mechanism underlying the effect of galectin on B cells, these technologies are applied in such a way that it is not clear what we learn beyond what is presented in the first two figures.

The authors provide evidence that the addition of gal-9 to B cells decreases BCR signaling at least to Erk (Fig. 3E). The question is: by what mechanism? The authors carry out pull-downs from B cell lysates using gal-9 beads, eluting bound material with lactose and analyzing the released material by mass spectrometry (Fig. 4). The authors claim that proteins identified in the eluate bind to gal-9. Unfortunately, this is simply not the case. Any protein that binds to something that binds to gal-9 will appear in the pull-down. Consequently, these data are not informative.

The authors proceed to use two color dSTORM imaging of gal-9 labeled or unlabeled B cells to describe the nanoscale organization of the BCRs and gal-9. Here the data analysis seems superficial, lacking controls, that leave the impression that little additional information is gained beyond that shown in the first two figures. To my knowledge, there are four reports on super resolution microscopy to analyze the spatial organization of BCRs on cell surfaces, Mattila et al. 2013; of which B. Treanor is a co-author, Avalos et al. 2014; Maity et al. 2015 and Lee et al. 2017. What seems to be apparent, at least from the Maity et al. and Lee et al. papers is that single molecule information can be obtained from super resolution images but only when single BCRs (and here gal-9) can be distinguished from multiple appearances of single molecules. It is not clear what proportion of either BCRs on gal-9 is accounted for in these images, how clusters are identified for BCRs and gal-9, what the relationship between small versus larger islands is, and what forces are predicted to influence these. It is not clear that the authors accounted for this potential pitfalls of the analysis.

These two problems will need to be addressed before the data provided are interpretable and add to our understanding of how galectin-9 affects B cells.

Reviewer #2 (Remarks to the Author):

The manuscript of Cao et al on Galectin-9 binds IgM-BCR to regulate mobility, clustering, and signaling describes the effect of the soluble lectin Galectin-9 (Gal-9) on the distribution and function of the B cell receptor (BCR). The immune modulation function of members of the Galectin-family is of increasing interest to the immunological community, but little is known on the function of Galectin on B cells. It recently has been shown that Gal-9 deficient mice have an exaggerated immune response and develop an auto-immunity. The aim of this study is to learn more about the effect of Gal-9 on the normal function of B-lymphocytes. For this, the authors, first study splenic B cells from Gal-9 deficient mice and compare them to their wild type (WT) counterpart. They find that only WT B-cells contain Gal-9 on their surface. They then use the confocal microscope to monitor how B-cells from these mice interact with an anti-BCR antibody coated surface and found that Gal-9 KO B cells display an increased spreading on such a surface. Furthermore, they show that Gal-9 KO cells show an increased production of phosphoERK (pERK).

They suggest that Gal-9 has a negative regulatory role on BCR signaling. In a complementary study they expose the WT and Gal-9 KO B cells to increasing amounts of Gal-9 and show that B cells exposed to Gal-9 have a reduced pERK response. They also used flag-tagged Gal-9 to purify Gal-9 binding proteins from Gal-9 KO B cells. With a ms. spec. analysis they identify the protein tyrosine phosphatase CD45, the IgM-BCR and its signaling subunit CD79b as Gal-9 binding molecules. An analysis of IgM-BCR nanoclusters on resting WT and Gal-9 KO B cells did not find a drastic nanostructures in these cells. However, they then find that exposure of the Gal-9 KO B cells to a large amount of Gal-9 and use the rearrangement of the IgM B cell nanoclusters. In a multicolored imaging study they also then find that the co-localization of areas on the B cell surface containing Gal-9 with that of CD22 IgM and CD45 suggesting that with the presence of Gal-9 the IgM BCR is in closer proximity to the CD45 and CD22 molecules and that may explain the inhibitory effect of Gal-9. Little is known of the effect on Gal-9 on B lymphocytes and as such the study is suitable for publication on Nature Communication. However, they have several points to clarify before I can recommend publication.

Major comments:

1. The authors utilize in figures and text the word "antigen" but in reality they always stimulate the BCR via anti-kappa or anti-IgM antibodies are a part. They should be aware that anti-receptor antibodies are a pure surrogate for the cognate antigen and this should correctly match the figures.
2. In their study of the effect of a Gal-9 deficiency on BCR signaling the author only tested for the production of pERK. The production of pERK is one of many events in BCR signaling. It would be good to show here also a total anti-gy Western as well as other more frequently used readouts such as total phosphatase in production and calcium response.
3. In figure 3 the authors also show that exposure of 1 μ M amounts of Gal-9 can inhibit the P-ERK response and conclude that Gal-9 is a negative regular for B lymphocytes. To support this thesis it would be very important to also show that exposure of B cells to 1 μ M of Gal-9 is not inducing any calcium response and the authors should also show this data. B cells exposed to anti-IgM antibodies should be used here as positive control.
4. In their sublementary Fig. S1 the authors show that Gal-9 B cells do not change the expression of IgM, IgD and CD19. What is missing is the source analysis for B cell exposed to 1 μ M of Gal-9 for different times.

Reviewer #3 (Remarks to the Author):

In this nice study Cao et al. investigate the role of Galectin 9 in BCR signalling. To this end, the authors take advantage of a variety of cutting-edge imaging techniques and combined them with genetics and biochemistry. This powerful combination allows the authors to conclude that Galectin 9 plays an important role in BCR signalling regulation by binding preferentially binding IgM and thereby altering its B cell surface organization and distribution. They also offered a plausible mechanism on how Galectin 9 performs this function by providing convincing evidence that the presence or absence of this lectin significantly alters the IgM association with key regulatory co-receptors CD22 and CD45. Of course, it will be interesting to know how this signalling alterations translate into in vivo B cell responses but these experiments are difficult given that Galectin 9 deficient animals will exhibit perturbations also in other immunological compartments, which will make impossible to dissociate from a B cell-intrinsic defect. In conclusion, the experiments presented here are well performed and the claims well supported. Therefore this study reveals novel aspects of BCR signalling control that are likely to carry important implications the understanding on how other receptors can be regulated.

Critics:

- 1) I will replace Porto 2004 as a reference in the introduction reference. This is outdated.
- 2) Similarly, too many references are used in the introduction and this needs to be concentrated in key studies

- 3) The introduction also requires a paragraph with some information about CD45 and CD22.
- 4) In figure 1 panels B and C quantifications should be added.
- 5) Figure 6 will benefit from some panels where individual tracks are added to give an idea of how much displacement is affected.
- 6) It will be helpful to have a schematic model on how Galectin 9 is altering the association of BCR with CD45 and CD22.
- 7) Finally, is the addition of Galectin 9 to Gal9KO sufficient to trigger

Revised manuscript no: NCOMMS-17-32945-T, entitled ‘Galectin-9 binds IgM-BCR to regulate mobility, clustering, and signaling by Anh Cao, Nouf Alluqmani, Fatima Hifza Mohammed Buhari, Laabiah Wasim, Logan K. Smith, Andrew T. Quaile, Michael Shannon, Zaki Hakim, Hossai Furmli, Dylan M. Owen, Alexei Savchenko and Bebhinn Treanor

Point by point reply to Referee #1 (R1)

R1 stated: *In this manuscript the authors explore the effect of the binding of the evolutionarily conserved lectin, galectin-9 (gal-9) on the spatial organization of IgM-containing B cell receptors (BCRs) and their signaling function. The role of galectin in modulating B cell responses is an area of genuine interest as glycan-binding proteins have been established to play a central role in the regulation of the immune system... Consequently, information on the mechanisms by which galectins function in B cell activation are of interest. Unfortunately, even though these expert authors use cutting edge biochemical and imaging techniques to address the mechanism underlying the effect of galectin on B cells, these technologies are applied in such a way that it is not clear what we learn beyond what is presented in the first two figures.*

We appreciate that R1 recognizes the importance of investigating the role of galectins in modulating B cell responses and our expertise in biochemical and imaging techniques. We thank the reviewer for their suggestions to improve our manuscript and have added new details of dSTORM analysis, additional experiments to verify our identification of galectin-9 ligands, as well as new data further investigating the mechanism for galectin-9 mediated suppression of BCR signaling.

R1 raised the following major points

1. The authors provide evidence that the addition of gal-9 to B cells decreases BCR signaling at least to Erk (Fig. 3E). The question is: by what mechanism? The authors carry out pull-downs from B cell lysates using gal-9 beads, eluting bound material with lactose and analyzing the released material by mass spectrometry (Fig. 4). The authors claim that proteins identified in the eluate bind to gal-9. Unfortunately, this is simply not the case. Any protein that binds to something that binds to gal-9 will appear in the pull-down.

We thank the reviewer for raising this concern and have addressed it in two new experiments. First, we have performed our pull-down followed by western blotting using more stringent lysing conditions to break up protein-protein interactions. We have used RIPA lysis buffer containing the detergents 1% Triton X-100, 0.1% sodium deoxycholate, and 0.1% SDS; conditions widely accepted to disrupt most protein-protein interactions. Under these conditions, we found CD45 and IgM in the eluates of lactose eluted flagged-tagged recombinant galectin-9. However, in contrast to our previous lysis conditions (mild detergent of 0.1 % Nonidet-P40), we could no longer detect CD79b – presumably because under our original conditions it was only associated with IgM as the reviewer pointed out. In addition, we included an additional condition in which lactose was added during the incubation with FLAG-tagged rGal9 to identify if the interaction between galectin-9 and the ligands were glycan-mediated. This new data has replaced Figure 4C (now Figure 4B as table was moved). Although these conditions should dissociate most protein-protein interactions, we sought an additional method to verify that galectin-9 binds directly to CD45 and IgM. To this end, we performed far western experiments, using cell lysates from primary murine B cells, the B cell line A20, which expresses endogenous IgG but not IgM, A20 B cells expressing Hel-specific IgM, and Jurkat T cells. Cell lysates were subjected to SDS-PAGE followed by re-naturation with decreasing concentrations of guanidine-HCl (6 M to 0 M) according to Wu et al. Nature Protocols,

2007. FLAG-tagged rGal9 was incubated with the membranes as bait protein followed by mouse anti-FLAG antibody and anti-mouse HRP. We detected two bands in these experiments in primary murine B cells and A20 B cells expressing Hel-IgM, which corresponded to the size of CD45 and IgM in reference blots. We did not detect the IgM band in A20 cells that do not express endogenous IgM, nor did we detect either CD45 or IgM bands in Jurkat T cells. This new data can be found in Figure 4C. Taken together, this new data strongly suggests that galectin-9 binds to CD45 and IgM. Further, we have shown specificity in galectin9-mediated clustering, as IgD and Fc γ RIIb are not enriched in galectin-9 high regions (Supplementary Figure 6).

2. The authors proceed to use two color dSTORM imaging of gal-9 labeled or unlabeled B cells to describe the nanoscale organization of the BCRs and gal-9. Here the data analysis seems superficial... What seems to be apparent, at least from the Maity et al. and Lee et al. papers is that single molecule information can be obtained from super resolution images but only when single BCRs (and here gal-9) can be distinguished from multiple appearances of single molecules. It is not clear what proportion of either BCRs on gal-9 is accounted for in these images, how clusters are identified for BCRs and gal-9, what the relationship between small versus larger islands is, and what forces are predicted to influence these.

We thank the reviewer for pointing out the brevity of our description of dSTORM analysis. We have extensively revised the description of our methodology for processing of dSTORM data using Thunderstorm (see pages 44-46) and added a supplementary figure to depict our workflow and key parameters of processing (Supplementary Figure 8). With regards to the multiple appearance of the same molecule, this is indeed a problem with dSTORM data as the reviewer points out. We understand that multiple appearances of single molecules (both in consecutive frames and random re-appearances throughout acquisition) can result in overcounting of BCR localizations and false positive BCR clusters that may in fact be BCR monomers. In Maity et al., consecutive localizations are accounted for such that localizations within 50nm, with a max “on” frame of 6 and max “off” frame of 10 are grouped. In Lee et al. consecutive localizations are also accounted for by collapsing localizations within 60nm radius to a single localization. Re-appearances were not accounted for in this study due to difficulties in the stochastic nature of the process, although they estimated 80% of labelled BCRs appear only once and the number of multiple occurrences is largest for 1 reappearance. The authors also showed that protein islands with fewer than 5 BCR single molecule localizations have a radius of 60nm, and therefore most likely localizations arising from the same BCR, while protein islands containing 5 or more BCRs and having a radius of >60nm represent BCR localizations from multiple Alexa Fluor 647-labeled BCR molecules.

Despite this analysis, the authors make clear that under saturating conditions of labelling BCRs, the number of single molecules detected may not be equal to the absolute number of BCRs but are an overestimate. And although overcounting will reduce the number of BCRs identified as monomers, the authors state that overcounting will not affect the evaluation of spatial organization and relative changes in clustering parameters, such as radius and density of protein islands, between conditions.

We address the issue of multiple localizations due to consecutive “on” frames and multiple reappearances in our post processing of dSTORM data. To our understanding, single labelled BCRs that fluoresce in multiple consecutive frames appear as clusters due to differences in localization in each frame as a result of changes in photon emission and localization uncertainty. Therefore, using ThunderSTORM, first we filter out any localizations with uncertainty greater than 20nm. We then group localizations that occur within a 20nm radius in an unlimited number of consecutive frames with 1 reappearance. We picked these parameters to maximize the F1 score in the performance evaluation module of ThunderSTORM, which compares single molecule localization data post-processing to simulated ground truth positions. Indeed,

using a merging radius of 60nm results in an increased number of false negatives as localizations from different BCRs within 60nm may be collapsed to a single molecule, particularly for highly clustered proteins such as CD45 and CD22. Also, 60nm radius of merging increased the number of false positives, where the calculated position of the merged molecules may not match ground truth positions within 20nm, which would affect our colocalization analysis (see table below; and Supplementary Figure 9).

	Radius							RMSE lateral [nm]
	[nm]	# of TP	# of FP	# of FN	precision	recall	F1-measure	
Simulated CD45 60nm merge	20	129214	45375	68404	0.74	0.654	0.694	9.318
Simulated CD45 20nm merge	20	147394	43213	50224	0.773	0.746	0.759	9.163
Simulated CD22 60nm merge	20	201519	55111	68308	0.785	0.747	0.766	9.152
Simulated CD22 20nm merge	20	209929	53672	59898	0.796	0.778	0.787	9.100
Simulated IgM 60nm merge	20	138010	27829	33972	0.832	0.802	0.817	9.000
Simulated IgM 20nm merge	20	140598	27230	31384	0.838	0.818	0.828	8.974

To further account for multiple re-appearances, we identify BCR clusters as a minimum of 5 localizations, consistent with the number of single BCR molecules predicted to inhabit the smallest protein island by Lee et al. Finally, we kept the post-processing parameters identical across our conditions and we agree with Lee et al. that over-counting will not hinder the comparison of relative changes between conditions. In interpretations of our results, we make statements on the ability of rGal9 to induce BCR clustering compared to WT and Gal9KO B cells, however we do not make statements of the proportion of BCRs that are monomers.

To address reviewer 1's subsequent question, we identify BCR clustering based on the Getis statistical analysis carried out using Supercluster (<http://stmc.unm.edu>) according to the following equation:

$$G_i(d) = \frac{\sum_{j=i}^n w_{ij}(d)x_j}{\sum_{j=1}^n x_j}, \text{ for } j \neq i$$

Where d is the pairwise distance between subregions i and j ; n is the number of subregions in the ROI; and w_{ij} a binary weight matrix where $w_{ij}(d)=1$ if the pairwise distance, d , between subregions i and j is less than the cutoff distance D_c and $w_{ij}(d) = 0$ if d is greater than D_c . This results in a ratio between the sum of the values of subregions within a distance d of the i^{th} subregion and the sum of the values of all subregions, which is a measure of local clustering.

In terms of what proportion of BCRs are accounted for, we label IgM BCRs under saturating conditions and image 8000-10000 frames. After post-processing we get an average of 5000 localizations in $(3 \mu\text{m} \times 3 \mu\text{m}) 9 \mu\text{m}^2$ and based on estimating number of BCRs by multiplication by a factor to equal the total

surface area of a primary B cell (79 μm^2) including a factor of 2 for membrane ruffling, we approximate 80 000 total IgM BCRs on the cell surface. This number falls within the previously reported number of total IgM BCRs; therefore, we assume we are visualizing a good proportion of actual BCRs in our analysis. However, again due to over-counting we do not make any statements on absolute number of BCRs visualized, but rather compare proportion clustered between conditions.

Finally, based on the evidence reported, we hypothesize that galectin-9 brings together preformed BCR nanoclusters to form larger clusters. This called into question what the relationship between smaller nanoclusters and larger clusters are in the context of galectin-9 mediated suppression of BCR signalling. In Lee et al., their results indicated that protein-protein interactions are likely to govern small, high density BCR nanoclusters, whereas such interactions are unlikely to govern larger BCR clusters in the case of antigen engagement as BCRs form larger but less dense clusters. Our results are different from BCR antigen engagement in that we observe rGal-9 increases the molecular density IgM BCRs in clusters but does not increase the proportion of clustered BCRs. Taken together we propose that galectin-9 mediates heterotypic protein-protein interactions via glycan binding between IgM and other molecular species, which we report are likely inhibitory receptors CD45 and CD22, however galectin-9 does not induce IgM homotypic interactions, consistent with no observed increase in the total proportion of clustered IgM. Therefore, our data challenges the dogma that clustering per se is indicative of signaling, and instead argues that it all depends on what's in the cluster – in this case, the data is consistent with galectin9-mediated increased density of IgM clusters together with CD45 and CD22. In contrast, in response to antigen, IgM form larger, less dense microclusters that may exclude inhibitory coreceptors and include stimulatory coreceptors.

Point by point reply to Referee #2 (R2)

R2 stated: *The manuscript of Cao et al on Galectin-9 binds IgM-BCR to regulate mobility, clustering, and signaling describes the effect of the soluble lectin Galectin-9 (Gal-9) on the distribution and function of the B cell receptor (BCR). The immune modulation function of members of the Galectin-family is of increasing interest to the immunological community, but little is known on the function of Galectin on B cells... Little is known of the effect on Gal-9 on B lymphocytes and as such the study is suitable for publication on Nature Communication.*

We thank the reviewer for their positive comments on our manuscript and the importance of elucidating the function of galectins in B cells.

R2 raised the following points

1. The authors utilize in figures and text the word "antigen" but in reality they always stimulate the BCR via anti-kappa or anti-IgM antibodies are a part. They should be aware that anti-receptor antibodies are a pure surrogate for the cognate antigen and this should correctly match the figures.

We thank the reviewer for this suggestion – we did not intend to imply that we were using cognate antigen but anti-BCR as a surrogate for antigen. We have added this clarification to the text and figures as suggested by R2.

2. In their study of the effect of a Gal-9 deficiency on BCR signaling the author only tested for the production of pERK. The production of pERK is one of many events in BCR signaling. It would be good to show here also a total anti-py Western as well as other more frequently used readouts such as total phosphatase in production and calcium response.

We thank the reviewer for the suggestion to extend our analysis of BCR signaling in Gal-9 deficient B cells. To this end, we have examined total tyrosine phosphorylation as suggested, as well as the phosphorylation of CD19 and Akt, as our other data suggested that this pathway might be particularly relevant in the context of galectin-9 regulation of BCR signaling. We have added this new data to Figure 2E - G. In addition, we have extended this analysis to treatment of cells with recombinant galectin-9 as a tool to explore galectin-9 mediated effects on BCR signaling. This new data has been added to Figure 3C-E.

3. In figure 3 the authors also show that exposure of 1 μ M amounts of Gal-9 can inhibit the P-ERK response and conclude that Gal-9 is a negative regulator for B lymphocytes. To support this thesis it would be very important to also show that exposure of B cells to 1 μ M of Gal-9 is not inducing any calcium response and the authors should also show this data. B cells exposed to anti-IgM antibodies should be used here as positive control.

We thank R2 for suggesting this important control experiment. We have conducted this experiment and find that addition of rGal9 does not induce calcium signaling in B cells. This new data has been added as Supplementary Figure 5.

4. In their supplementary Fig. S1 the authors show that Gal-9 B cells do not change the expression of IgM, IgD and CD19. What is missing is the source analysis for B cell exposed to 1 μ M of Gal-9 for different times.

As suggested by R2, we have treated B cells with 1 μ M recombinant Gal-9 as in our other experiments (signaling, bilayers, SPT, dSTORM) and examined surface expression of IgM, IgD, CD19, and CD45. We find that treatment with rGal9 does not affect surface expression of IgM, IgD, CD19, or CD45. This new data is presented in Supplementary Figure 3.

Point by point reply to Referee #3 (R3)

R3 stated: *In this nice study Cao et al. investigate the role of Galectin 9 in BCR signalling. To this end, the authors take advantage of a variety of cutting-edge imaging techniques and combined them with genetics and biochemistry. This powerful combination allows the authors to conclude that Galectin 9 plays an important role in BCR signalling regulation by binding preferentially binding IgM and thereby altering its B cell surface organization and distribution. They also offered a plausible mechanism on how Galectin 9 performs this function by providing convincing evidence that the presence or absence of this lectin significantly alters the IgM association with key regulatory co-receptors CD22 and CD45... the experiments presented here are well performed and the claims well supported. Therefore this study reveals novel aspects of BCR signalling control that are likely to carry important implications the understanding on how other receptors can be regulated.*

We thank the reviewer for their recognition of our use of ‘a variety of cutting-edge imaging techniques combined with genetics and biochemistry’ to provide ‘compelling evidence’ that galectin-9 alters the association of IgM with CD22 and CD45. We agree with R3 that our study ‘reveals novel aspects of BCR signaling control’.

R3 raised the major points

1) I will replace Porto 2004 as a reference in the introduction reference. This is outdated.

Fair point – this reference has been updated. Thank you.

2) *Similarly, too many references are used in the introduction and this needs to be concentrated in key studies.*

We thank the reviewer for this suggestion and have focused our references on a few key studies. We do note, however, that the concepts discussed in the introduction span a variety of topics in different cell types and different conditions and therefore have tried to represent this in our choice of references – as we are not over the limit of references we have still elected to provide the reader with a number of papers to refer to on these topics.

3) *The introduction also requires a paragraph with some information about CD45 and CD22.*

Thank you for this suggestion – given word limit constraints and the flow of the text we have included some background information on CD45 and CD22 at relevant places in the results section (see page 13, 14).

4) *In figure 1 panels B and C quantifications should be added.*

Thank you for this suggestion – we have added quantification of our flow cytometry and confocal imaging data to Figure 1.

5) *Figure 6 will benefit from some panels where individual tracks are added to give an idea of how much displacement is affected.*

Again, this is a good suggestion – we have added panels to Figure 6C showing examples of individual tracks inside and outside of galectin-9 rich regions.

6) *It will be helpful to have a schematic model on how Galectin 9 is altering the association of BCR with CD45 and CD22.*

We have added a schematic model of how galectin-9 may alter that association of BCR with CD45 and CD22 based on current data. This model is presented in Figure 10.

7) *Finally, is the addition of Galectin 9 to Gal9KO sufficient to trigger.*

We thank R3 for suggesting this important control experiment. We have conducted this experiment and find that addition of rGal9 does not induce calcium signaling in B cells. This new data has been added as Supplementary Figure. 5.

REVIEWERS' COMMENTS:

Reviewer #1 (Remarks to the Author):

The authors have addressed all of my concerns by the addition of new data and editing.

Reviewer #3 (Remarks to the Author):

The authors have done an excellent job to address my initial concerns. As an expert in B cell Biology and imaging I do believe that this work is an important contribution to the field.